# DKK3 Expression in Glioblastoma: Correlations with Biomolecular Markers

**DOI:** 10.3390/ijms25074091

**Published:** 2024-04-07

**Authors:** Maria Caffo, Giovanna Casili, Gerardo Caruso, Valeria Barresi, Michela Campolo, Irene Paterniti, Letteria Minutoli, Tamara Ius, Emanuela Esposito

**Affiliations:** 1Unit of Neurosurgery, Department of Biomedical and Dental Sciences and Morphofunctional Imaging, University of Messina, 98100 Messina, Italy; mcaffo@unime.it; 2Department of Chemical, Biological, Pharmaceutical and Environmental Sciences, University of Messina, 98100 Messina, Italy; giovanna.casili@unime.it (G.C.); michela.campolo@unime.it (M.C.); irene.paterniti@unime.it (I.P.); emanuela.esposito@unime.it (E.E.); 3Department of Diagnostics and Public Health, Section of Pathology, University of Verona, 37124 Verona, Italy; valeria.barresi@univr.it; 4Department of Clinical and Experimental Medicine, University of Messina, 98100 Messina, Italy; letteria.minutoli@unime.it; 5Neurosurgery Unit, Head-Neck and NeuroScience Department, University Hospital of Udine, 33100 Udine, Italy; tamara.ius@gmail.com

**Keywords:** DKK3, glioblastoma, glioma, IDH status, survival, WNT pathway

## Abstract

Glioblastoma is the most common malignant primary tumor of the CNS. The prognosis is dismal, with a median survival of 15 months. Surgical treatment followed by adjuvant therapies such as radiotherapy and chemotherapy characterize the classical strategy. The WNT pathway plays a key role in cellular proliferation, differentiation, and invasion. The DKK3 protein, capable of acting as a tumor suppressor, also appears to be able to modulate the WNT pathway. We performed, in a series of 40 patients, immunohistochemical and Western blot evaluations of DKK3 to better understand how the expression of this protein can influence clinical behavior. We used a statistical analysis, with correlations between the expression of DKK3 and overall survival, age, sex, Ki-67, p53, and MGMT and IDH status. We also correlated our data with information included in the cBioPortal database. In our analyses, DKK3 expression, in both immunohistochemistry and Western blot analyses, was reduced or absent in many cases, showing downregulation. To date, no clinical study exists in the literature that reports a potential correlation between IDH and MGMT status and the WNT pathway through the expression of DKK3. Modulation of this pathway through the expression of DKK3 could represent a new tailored therapeutic strategy in the treatment of glioblastoma.

## 1. Introduction

According to the most recent classification of the World Health Organization (WHO), diffuse gliomas in adults are classified into glioblastoma (GB) IDH-wild-type, astrocytoma IDH-mutant and oligodendroglioma IDH-mutant, and 1p/19q codeleted [1].

Glioblastoma IDH-wild-type is the most common malignant primary tumor in the central nervous system (CNS), accounting for 14.5% of all CNS tumors and 48% of malignant CNS tumors [2]. GB shows a dismal prognosis, with a reported median overall survival (OS) of approximately 15 months [3]. Surgical treatment followed by radiotherapy and chemotherapy represent the standard of care for patients harboring this tumor. Improvements in surgical techniques, including intra-operative mapping of eloquent areas of the brain and the use of fluorescent dyes that are helpful in the detection of tumor borders, show only a slight increase in survival [4,5]. GB differs at the genetic and epigenetic levels, especially in discriminating the mutational status of the metabolic enzyme isocitrate dehydrogenase 1 (IDH1), which serves as a prognostic factor [6]. Numerous studies show that IDH1 mutations lead to better OS in patients and a better response to therapies [7,8]. 

The WNT pathway plays key roles in the cellular proliferation, differentiation, and invasion of gliomas [9,10]. The WNT pathway can be regulated by Dickkopf (*DKK*) proteins, including the *DKK3*-related protein, termed *Soggy* (*DKKL1* or *SGY-1*). In particular, DKK3 overexpression is able to suppress tumor cell growth and seems to be the most promising tumor suppressor molecule able to modulate the WNT pathway [11,12]. WNT signaling is aberrantly activated in GB, inducing growth and invasion [13,14]. Very few studies have investigated the relationship between IDH1 mutation and WNT signaling in GB IDH-wild-type, especially concerning DKK3 expression. IDH1 mutation is associated with reductions in cell survival, proliferation, and the invasion of GB by decreasing WNT signaling [15,16]. In addition, the activation of the canonical WNT/β-catenin signaling cascade induces the expression of DNA repair enzyme O6-methylguanine-DNA methyltransferase (MGMT) expression, a ubiquitously expressed enzyme that is commonly overexpressed in GB which is associated with resistance to alkylating agents. MGMT is regulated by multiple mechanisms, including epigenetic silencing of the MGMT gene by promoter methylation, and is implicated in the development of chemoresistance [17].

In our previous study, we investigated “in vitro” and “in vivo” the biological role of DKK3 in tumor cells of GB, finding that a modulation of this protein could promote apoptosis, providing a potential strategy for the treatment of GB [18]. Particularly, significant decreases in DKK3 were observed in human glioblastoma cell lines, as well as in U-87 MG xenograft tumors and in GB human patients’ tissues, highlighting that a combined modulation of the WNT/DKK3 pathways, simultaneously targeting apoptosis and survival signaling defects, might shift the balance from tumor growth stasis to cytotoxic therapeutic responses, resulting in greater therapeutic benefits. Here, we analyze the potential prognostic relevance of DKK3 expression, with immunohistochemistry and Western blot, and its correlation with molecular and histopathological features in 40 GB. In addition, we assess possible correlations between the expression of DKK3 and overall survival, age, gender, Ki-67, p53, MGMT, and IDH status with statistical analysis. Validation of our data was also considered, correlating our results with the cBio Cancer Genomics Portal (cBioPortal) database and also creating a virtual study using data provided by previous studies and by samples from 2220 GB patients [19]. To date, no study exists in the literature that reports a potential correlation between IDH and MGMT status and the WNT pathway through the expression of DKK3. 

## 2. Results

This series included 40 patients (24 women and 16 men) aged between 43 and 77 years with an average age of 53 years. All patients underwent surgical treatment. In Table 1, we summarize the data relating to the location of the tumor as well as the symptoms presented by the patients. At MR with gadolinium, the lesion appears as a contrast-enhancing mass, with a thickened ring of enhancement and a hypointense core, which usually corresponds to central areas of necrosis. The margins of the tumor may be irregular or poorly defined, sometimes with spread along white matter tracts. MR also detects the degree of edema surrounding the tumor, and susceptibility imaging can show whether the tumor contains micro-hemorrhages. A biopsy was performed on two patients, in which the tumor was deeply located. In these patients, we adopted a neuro-navigation system to guide the biopsy needle. In the remaining cases, a standard craniotomy, under general anesthesia, was performed. We used intra-operative navigation in all patients to better determine the tumor location and plan the incision and craniotomy accordingly. In addition, we performed fluorescence-guided surgery that increased and improved the extent of resection (EOR). When maximum resection is not feasible (90% of resection), supratotal resection of the tumor is an option, which is defined as resection beyond tumor mass enhancement showed by MR. In our patients, we achieved maximum resection in twelve cases; in the remaining patients we obtained supratotal resection. Intercellular areas of necrosis, endothelial hyperplasia, and mitotic figures were evidenced in all specimens. The majority of cells showed small dark nuclei and multiple fibrillary processes. Pseudopalisading was also observed in a large number of cases. In all cases, biomolecular studies were performed, as previously described. In thirty-two cases, the lesion was GB IDH-wild-type, while in the remaining eight it was a grade IV astrocytoma IDH-mutant. Fifteen patients are still alive; twenty-five died. One patient died from pneumonia during chemotherapy, another patient died from cerebral empyema, and two more died of acute abdomen from intestinal obstruction. The remaining patients died from disease progression. Thirty-nine patients underwent the Stupp protocol with radiotherapy and chemotherapy with temozolomide. All patients with disease progression supported by MR with gadolinium and flair sequences, according to the international RANO criteria, underwent two lines of chemotherapy, with nitrosuree (fotoemustin) and/or bevacizumab or regorafenib. The duration of follow-up ranged from 2 to 60 months. Every two months, all patients underwent MR to evaluate their response to treatments. The live patients are still in follow-up. 

### 2.1. Expression of DKK3

The expression of DKK3 was evaluated using immunohistochemistry (Figure 1) and Western blot analyses (Figure 2) and was graded based on the positive ratio and the intensity of the immunoreaction,. Thirty-eight cases were positive. The ID score was nine in 22 cases, six in 12 cases, and one in 4 cases. Two cases did not show immunoreactivity; therefore, they had a score of 0. Positive immunostaining was mostly observed in the cytoplasm of endothelial neoplastic cells. In addition, we found that normal brain tissue contained many diffusely distributed DKK3-positive cells. 

### 2.2. DKK3 Expression and Sex/Age

Our data highlighted that a low expression of DKK3 was more frequent in male patients, while in female patients, a higher expression of DKK3 was more frequent (Figure 3). In our research, we observed a significant correlation between an increase in age and absent or lower DKK3 expression, observing absent or lower expression of DKK3 in patients with an average age of 57 years (Figure 4). 

### 2.3. DKK3 Expression and Survival

The analysis between DKK3 expression and overall survival after the first surgery for GB did not show a statistically significant correlation, although an increase in survival was observed in patients with moderate and high DKK3 expression (28 months) compared to a median of 15 months of survival in patients with lower or absent expression of DKK3 (Figure 5). 

### 2.4. DKK3 Expression and Ki-67 Labeling Index

The median of the Ki-67 labeling index was lower in patients with moderate (23%) and high expressions of DKK3 (28.8%) compared to 43% in patients with absent expression of DKK3 (Figure 6). Interestingly, we noted that in our cases, 17 cases had Ki-67 levels higher than 25%, 3 cases were >80%, and 7 cases had Ki-67 values >50%; of these latter cases, four cases had a survival >11 months, and two cases showed a survival of 19 months. The three cases that displayed Ki-67 values >80% showed a survival, respectively, of 1 month, (died of cerebral empyema), 23 months, and 18 months. 

### 2.5. DKK-3 Expression and IDH Status

In this study, patients with IDH-wild-type and patients with IDH-mutant showed a moderate and high expression of Dkk3 compared to the remaining patients (six) with absent or low DKK3 expression (Figure 7A). To better understand the impact of IDH-wild-type or -mutant on survival, related to DKK3 expression, we performed a correlation using these three parameters. Interestingly, an increased survival, expressed in months, was observed in patients with moderate and high DKK3 expression, both with IDH-wild-type and -mutant (Figure 7B). 

### 2.6. DKK3 Expression and MGMT Methylation Status

Patients with moderate and high expressions of DKK3 showed a significant reduction in the percent of methylation (11.9%) of the MGMT gene compared to patients with absent or low DKK3 expression (26.3% and 26.8%, respectively) (Figure 8). 

### 2.7. DKK3 Expression and p53

Patients with moderate and high expressions of DKK3 presented a significant, elevated, detectable p53 expression compared to patients with low or absent DKK3 expression (Figure 9). 

### 2.8. Correlations with cBioportal Database

The correlations were performed using cBioportal for cancer genomics, querying 2241 samples from 2220 GB patients’ cases (Figure 10). The analysis performed showed a prevalence of pathology with 1152 cases registered in men compared to women (740), supporting our clinical data that a low expression of DKK3 tends to be higher in male patients compared to female patients. In detail, the data obtained were systematized as a function of patient and gene, and the portal’s fundamental construct is the concept of altered genes; specifically, the analysis classifies a gene as altered in a specific patient if it is mutated, homozygously deleted, amplified, or if its relative mRNA expression is less than or greater than a user-defined threshold. In light of these observations, a range of incidence from 10 to 89 years was evidenced, with an increase in frequency from 30 years in patients with unaltered DKK3. Interestingly, in patients with mutated DKK3 (altered group), the incidence age ranged from 25 to 80 years, suggesting an involvement of DKK3 in postponing disease onset. The performed analysis showed an alteration of DKK3 in only 13 patients, compared to 2015 patients which presented an unaltered (not mutated) *DKK3* gene. Interestingly, the median months of survival in the altered DKK3 group was higher (21.9 months) compared to 14.9 months of survival observed in patients with unaltered DKK3, suggesting a protective effect of DKK3 on survival (Figure 11). A statistical analysis was performed using the log-rank test as a hypothesis test to compare the survival distributions of samples. It is a nonparametric test and appropriate to use when the data are right-skewed and censored. In GB, IDH1 R132H occurs at the active site of the enzyme and at the highly conserved codon 132 and causes Arg to His substitution. The analysis performed on 2220 patients highlighted IDH1 mutations in the mutated-*DKK3*-gene group (altered group). Specifically, 12% of the samples showed R132H mutations. In fact, R132C and R132G IDH1 mutation results appeared less common. Particularly, this analysis highlighted 50% of R132G mutations in GB patients compared to IDH1-wild-type observed in the group without DKK3 mutations. The analysis performed on 2220 patients revealed an increase MGMT methylation in the mutated DKK3 group (23%) compared to 8% of methylation status in the group without DKK3 mutations. Statistical analysis of cBioPortal data was performed using the Kruskal–Wallis test as nonparametric equivalent to the ANOVA test.

## 3. Material and Methods

Our study included 40 patients (32 GB-IDH-wild-type and 8 astrocytoma IDH-mutant). All patients provided informed consent and tumor collection was approved by the local Ethical Board. For each case, we collected the following data: age, sex, clinical presentation, tumor location, extent of surgical resection, immunohistochemical (IDH1, p53, and Ki-67) and biomolecular features, adjuvant therapies, and survival. Table 2 shows the biomolecular analyses and survival of each patient. The extent of surgical resection was classified as biopsy, subtotal, total, or supratotal resection. 

### 3.1. Immunohistochemistry

Tissue samples obtained from surgical operation were immediately stored in a deep freezer at −70 °C. All samples were formalin-fixed for 24 h at room temperature, paraffin-embedded at 56 °C, and cut into parallel 4 µm thick sections for histological evaluation with hematoxylin and eosin stain (H&E) and for immunohistochemical analyses against DKK3. The intrinsic endogenous peroxidase activity was blocked with 0.1% H_2_O_2_ in methanol for 20 min; then, normal sheep serum was applied for 30 min to prevent unspecific adherence of serum proteins. DKK3 antigens were unmasked by microwave oven pre-treatment in 10 mM, pH 6.0 sodium citrate buffer for 3 cycles × 5 min. Consecutive sections were successively incubated with the primary antibodies against DKK3 (Abcam, Cambridge, UK; working dilution, 1:100) by using a Dako Autostainer. The bound primary antibodies were visualized by using an envision system (Dako Cytomation, Glostrup, Denmark) according to the manufacturer’s instructions. To reveal the immunostaining, the sections were incubated in the dark for 10 min, with 3-3′ diaminobenzidine tetra hydrochloride (Sigma Chemical Co., St. Louis, MO, USA) in the amount of 100 mg in 200 mL 0.03% hydrogen peroxide in phosphate-buffered saline solution (PBS). Nuclear counterstaining was performed by Mayer’s hemalum. The immunohistochemical expression of DKK3 was assessed semi-quantitatively, by using the so-called Intensity Distribution (ID) score. In detail, in all cases the intensity of staining (IS) was classified as 0 (absent), 1 (weak), 2 (moderate), 3 (strong). The area of positive staining (ASP), which reflected the percentage of positive cells, was rated as follows: 0 (less than 5% of stained cells), 1 (5–25% stained cells), 2 (26–50% stained cells), 3 (51–75% stained cells), 4 (>75% stained cells). Then, for each case the ID score was calculated by multiplying the values of IS and ASP. All cases were immunostained using antibodies against IDH1-R132H (clone H09, Dianova, Gmbh, Germany; dilution 1:200), p53 (clone DO-7, Leica Biosystems, Newcastle, UK; prediluted), and ATRX (Polyclonal; Life Science Sigma, St Louis, MO, USA; dilution 1:750) through an automated Immunostainer (Leica Biosystems, Newcastle, UK). The assessment and scoring of p53-stained slides were performed as previously suggested [20].

### 3.2. Western Blotting Analysis

Total cellular proteins of each specimen were extracted as previously described [18]. The obtained lysates were centrifuged at 13,000 rpm for 15 min and the supernatant was used for the determination of protein concentration by a Bio-Rad protein assay (BioRad, Richmond, CA, USA) and diluted with Laemmli buffer. GB specimens were denatured, separated by electrophoresis on SDS polyacrylamide gel, and then transferred onto nitrocellulose membrane at 200 mA for 1 h. The membranes were blocked with 5% non-fat dry milk and incubated with a primary antibody for Dkk3 (1:500, Abcam, Cambridge, UK) overnight at 4 °C. The day after, the membranes were incubated with a specific peroxidase-conjugated secondary antibody (Pierce, Cramlington, UK) for 1 h at room temperature and were analyzed by enhanced chemiluminescence (KPL, Milford, MA, USA). Protein signals were quantified by scanning densitometry using a bio-image analysis system (Bio-Profil, Milan, Italy) and the results were expressed as the relative integrated intensity compared to controls. GAPDH (1:500, Santa Cruz Biotechnology, Dallas, TX, USA) was used to confirm equal protein loading and blotting. Signals were detected with enhanced chemiluminescence (ECL) detection system reagent according to the manufacturer’s instructions (Thermo, Waltham, MA, USA). The relative expression of the protein bands was quantified by densitometry with BIORAD ChemiDocTMXRS+software. (Version 3.0.1) Images of blot signals (8 bit/600 dpi resolution) were imported to analysis software (Image Quant TL, v2003). Protein signals were quantified by scanning densitometry using a bio-image analysis system (Bio-Profil, Milan, Italy) and the results were expressed as the relative integrated intensity compared to controls.

### 3.3. Biomolecular Analyses

The mutational status of IDH1 and IDH2 was evaluated using a Mutation Detection Kit (Entrogen, Woodland Hills, CA, USA) and real-time PCR amplification. We were able to identify 9 mutations of IDH 1 (R100Q, R132H, R132C, R132S, R132G, R132L, R132P, R132V, R132X) and 9 mutations of IDH2 (R172K, R172M, R172W, R172G, R172S, R140Q, R140W, R140L, R140G). We also determined the methylation status of the MGMT promoter using pyrosequencing and a therascreen kit MGMT Pyro (Qiagen, Hilden, Germany). MGMT promoter methylation was classified as “no methylation” (0–9%), “low methylation” (10–29%), or “high methylation” (30–100%), as previously reported [21].

### 3.4. Statistical Analysis

DKK3 expression was correlated with the sex, age, and overall survival of the patients and with IDH mutational status, MGMT promoter methylation, p53 expression, and the Ki-67 labeling index. The extent of DKK3 expression in each case was evaluated by an arbitrary scale system; this scale divides the expression of DKK3 expression in a scale with grades 0–4: grade 0: absence of DKK3 expression; grade 1: low expression; grade 2: moderate expression; grade 3: high expression.

Results are presented as mean values ± standard error of the mean (SEM) of at least three independent experiments or representative Western blots of at least three independent determinations, unless indicated otherwise. Statistical analyses were performed with GraphPad Prism 8 (GraphPad Software; San Diego, CA, USA), using Student’s *t*-test, One-Way ANOVA or Two-Way ANOVA with multiple comparison and Bonferroni post hoc test where appropriate. *p*-values < 0.05 were considered statistically significant. 

### 3.5. cBioPortal Database and Bioinformatics Methods

cBioPortal (https://www.cbioportal.org/ accessed on 16 April 2021) is a free database for exploring, evaluating, and analyzing multidimensional cancer genomics data [22]. This database enables large-scale data processing, statistical analysis, and graphical review of tumor variations from gene to protein level. The portal contains datasets of a large number of cancer studies, including Cancer Cell Line Encyclopedia (CCLE) and Cancer Genome Atlas (TCGA, https://www.cancer.gov/tcga, accessed on 16 April 2021). We also based this research on information generated by the TCGA research network [19,22]. The research was carried out, as a virtual study, by using a combination of data, querying 2241 samples from 2220 GB patients’ cases. All cBioPortal data had the same clinical criteria and equally managed and normalized data, allowing comparative analyses of samples between different studies. We utilized the OQL programming language (Onco Query Language), and interrogated for the *DKK3* gene and the parameters selected for our analysis. In detail, the correlations were performed between the *DKK3* gene in GB samples and sex, age, survival, and IDH and MGTMT status. Data obtained from cBioPortal database do not require ethical approval. All patients whose samples were used in this analysis signed informed consent.

## 4. Discussion

The prognosis of patients with GB remains extremely poor, despite important improvements in surgery and in radio- and chemotherapies. Improvements in neurosurgical techniques, including intra-operative mapping of the eloquent brain areas and fluorescence-guided resection, and in new intra-operative neuroradiological tools can extend survival and ameliorate outcomes [4,5]. Surgery allows for histologic confirmation of the diagnosis as well as cytoreduction, may also provide a therapeutic role by decreasing intracranial pressure, and, sometimes, leads to the recovery of focal neurological function. Applications of biotechnology and improvements in methods of targeted delivery have led to some extension of the length of overall survival. 

Recent literature data support the role of the WNT pathway in glioma proliferation and invasion [9,10], suggesting that the inhibition of this pathway could represent a potential approach to cancer therapies. DKK3 shows the ability to modulate the activation of WNT receptors [23,24]. It has also been reported that DKK3 expression is essentially absent in GB and significantly reduced in glioma cell lines. In our analyses, DKK3 expression in immunohistochemistry and Western blot analyses was reduced or absent in many cases. These data confirm our previous results, suggesting the role of DKK3 in promoting tumor growth due to its loss of function as a tumor suppression gene [18]. Intriguingly, in our results, increased survival was observed in patients with a moderate and high DKK3 expression (28 months) compared to a median of 15 months of survival in patients with low or absent expression of DKK3. An increase in DKK3 expression could lead to a reduction in the cell survival, proliferation, and invasion of glioma. 

Our statistical analysis, concerning DKK3 expression and sex incidence, underlines that a low expression of DKK3 was found in a large number of male patients, compared to a higher expression of DKK3 found in female patients. This result was confirmed by information obtained using cBioPortal for cancer genomics. According to the literature, we can hypothesize a protective role of the estrogen receptor, which is already described as a tumor suppressor in GB. Furthermore, progesterone inhibits the glycolytic metabolism in GB as well as EGFR/PI3K/Akt/mTOR signaling [25], which are highly active in GB and strictly correlated to the WNT pathway, as we previously demonstrated [18,26]. Overall, sex differences in GB remain largely unknown, and clinical trials have never investigated them.

The incidence of GB increases with age, reaching its peak at 75–84 years [3]. We observed an absent or lower expression of DKK3 in patients with an average age of 57 years. Interestingly, we demonstrated a significant statistical correlation between an increase in age and absent or lower DKK3 expression. 

Ki-67 is a non-histone nuclear protein that is expressed by cells entering the mitotic cycle, whose extent of expression is roughly proportional to the GB histologic grade [27]. It has recently been reported that a higher proliferation index may be more susceptible to adjuvant therapy [28]. In the same manner, a positive correlation between the Ki-67 labeling index and overall survival in GB patients has been confirmed [27,29]. Our results support a relationship between DKK3 expression and cellular proliferation, and their correlation could be used in further independent study. 

Major advances in cancer genetics have revealed that the genes encoding IDHs are frequently mutated in a variety of human malignancies, including GB [30,31]. More than 90% of the mutations in IDH1 are R132H, which has been associated with significantly improved prognosis and longer progression-free and overall survival [32,33]. IDH1-R132H mutations lead to both a less aggressive phenotype and radiosensitization of glioma cells [34]. For the first time, in this study, we correlated IDH status with Dkk-3 expression. Our results show a moderate and high expression of DKK3 in 34 patients (both wild-type and IDH-mutated) compared to the remaining patients (6) with absent or low DKK3 expression. In addition, to better understand the impact of IDH status (wild-type or mutated) on survival, related to DKK3 expression, we performed a correlation among these three parameters. Interestingly, an increased overall survival was observed in patients with moderate and high DKK3 expression, both with IDH-wild-type and -mutant (Figure 7). Although these data are not statistically significant, a higher expression of DKK3, associated with a mutated IDH status, could lead to an increase in survival.

It has been recently reported that the inhibition of the WNT pathway downregulates MGMT expression and re-establishes the chemosensitivity of DNA-alkylating drugs in mouse models [17]. Interestingly, in our study, patients with moderate and high expressions of DKK3 showed a reduction in the percent of methylation of the MGMT gene, compared to patients with absent or low DKK3 expression, also revealing that cases showing a higher percentage of methylation had a survival of 10 months. Therefore, for the first time, we showed that the correlation between DKK3 and MGMT could have a prognostic role.

p53 plays a central role in maintaining cellular homeostasis and is frequently deregulated in cancer [35]. For the first time, in our study, we demonstrated that GB patients with moderate and high expressions of DKK3 were correlated with a significant elevated detectable p53 expression compared to patients with absent DKK3 expression, supporting our hypothesis that DKK3 can be considered a prognostic factor in GB.

The results of our study can lay the foundations for future studies on a large number of patients to confirm role of DKK3 in GB initiation and progression. Additionally, WNT pathway activity greatly contributes to resistance to standard chemotherapy; elements of this pathway appear as attractive supplementary treatment targets both in early and progressive GB. In addition, new mutations or proteins expressed in GB offer the potential for the design of new compounds with tailored targets, and the association of mutations within tumors with different clinical courses facilitates the diagnosis and prediction of disease gravity and prognosis. Furthermore, the development of advanced computational and statistical methodologies, such as cancer genome data from large-scale projects, significantly contribute to developing personalized targets and biomarkers and to establishing the clinical relevance of cancer genomic pathway discovery. 

## Figures and Tables

**Figure 1 ijms-25-04091-f001:**
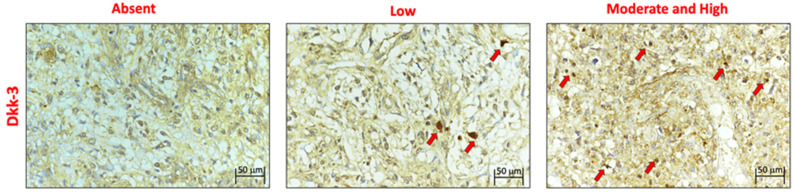
Immunohistochemical expression of DKK3 (red arrows indicate positive neoplastic endothelial cells).

**Figure 2 ijms-25-04091-f002:**
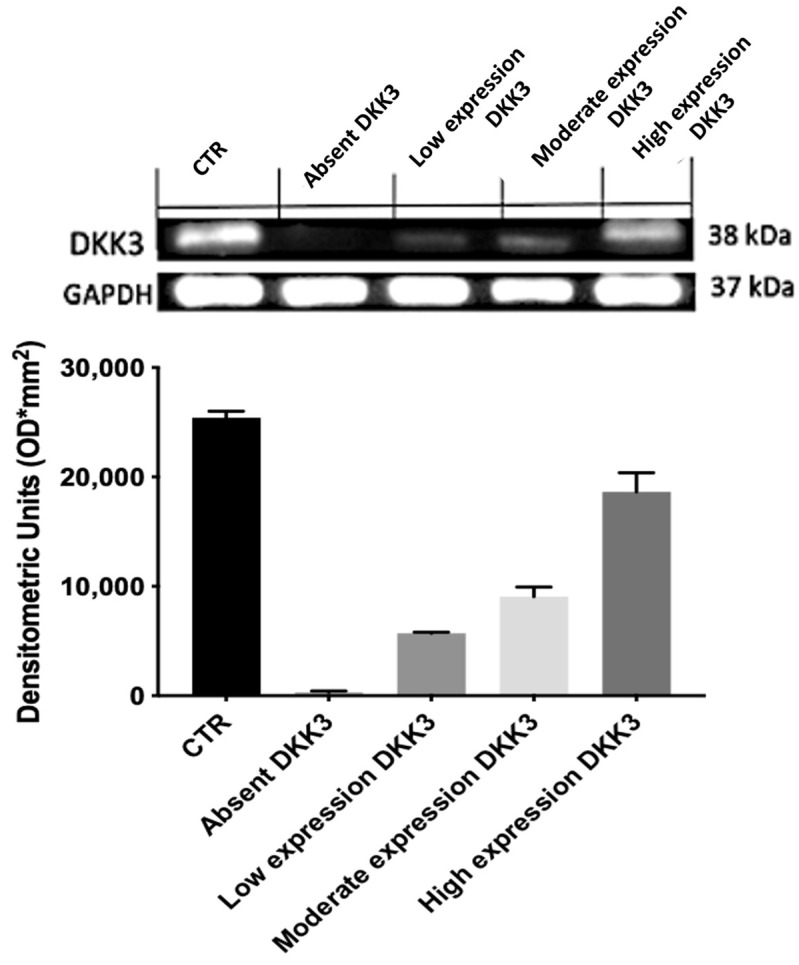
DKK3 Western blot analyses.

**Figure 3 ijms-25-04091-f003:**
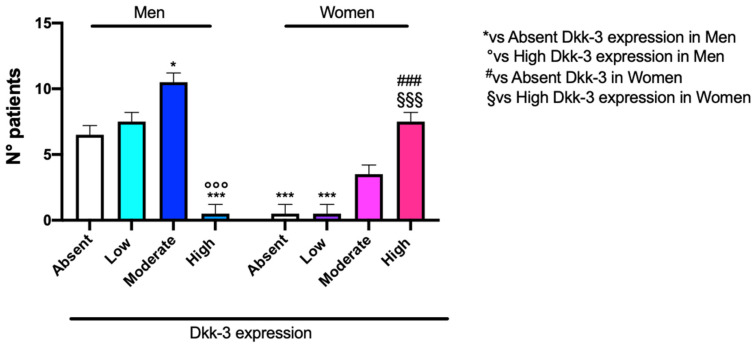
The correlation between DKK3 expression and sex. Two-Way ANOVA followed by Bonferroni post hoc test for multiple comparisons; * *p* < 0.05 and *** *p* < 0.001 vs. absent DKK3 expression in men; °°° *p* < 0.001 vs. high DKK3 expression in men; ### *p* < 0.001 vs. absent DKK3 expression in women; §§§ *p* < 0.001 vs. high DKK3 expression in women. Data refer to mean values of N = 3 independent experiments ± SEM.

**Figure 4 ijms-25-04091-f004:**
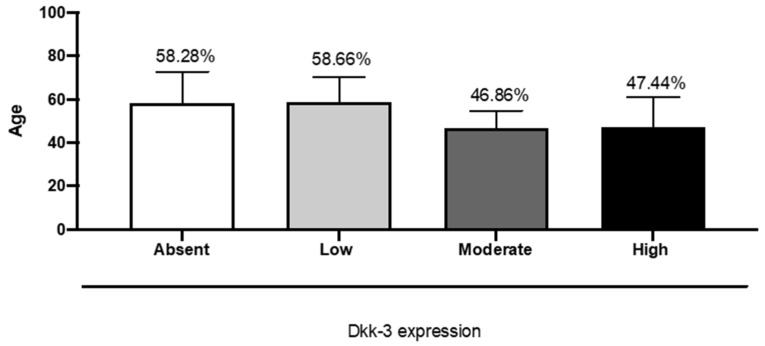
The correlation between DKK3 expression and age. One-Way ANOVA followed by Bonferroni post hoc test for multiple comparisons. Data refer to mean values of N = 3 independent experiments ± SEM. F value = 3.22; *p* value = 0.03; *p* value summary = *; R squared = 0.212.

**Figure 5 ijms-25-04091-f005:**
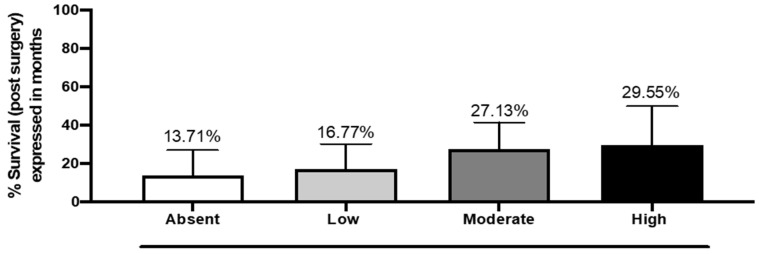
The correlation between DKK3 expression and survival. One-Way ANOVA followed by Bonferroni post hoc test for multiple comparisons. Data refer to mean values of N = 3 independent experiments ± SEM. F value = 2.24; *p* value = 0.10; *p* value summary = ns; R squared = 0.157.

**Figure 6 ijms-25-04091-f006:**
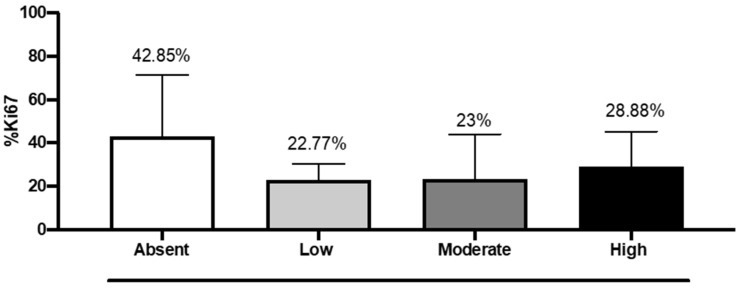
The correlation between DKK3 expression and % Ki-67. One-Way ANOVA followed by Bonferroni post hoc test for multiple comparisons. Data refer to mean values of N = 3 independent experiments ± SEM. F value = 1.90; *p* value = 0.15; *p* value summary = ns; R squared = 0.137.

**Figure 7 ijms-25-04091-f007:**
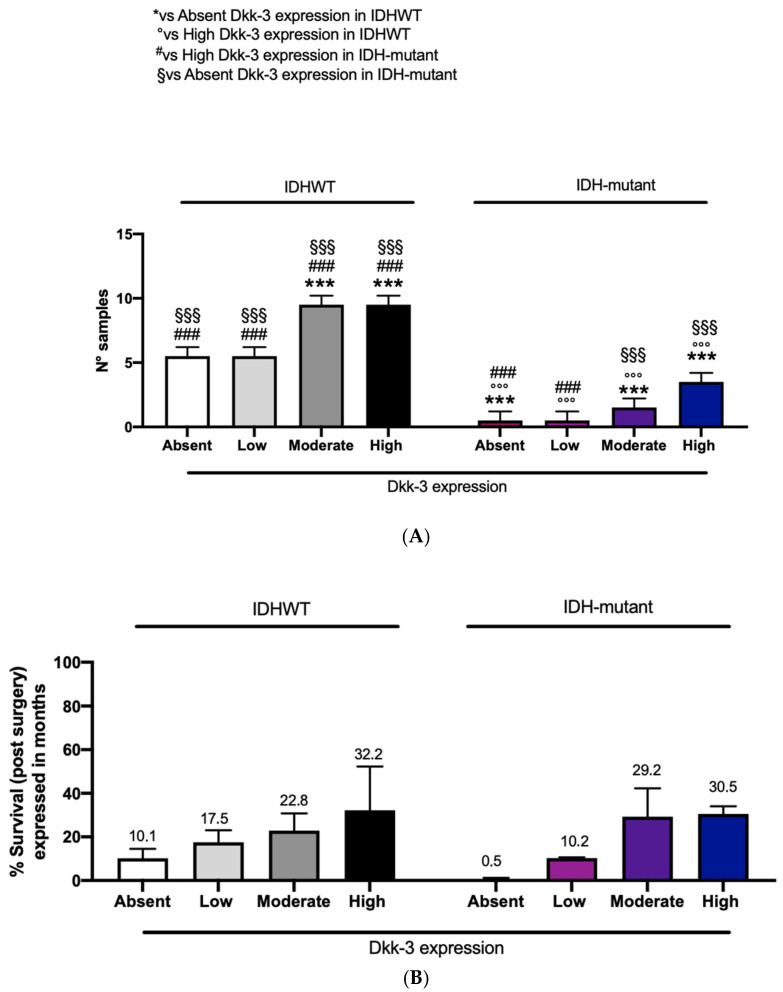
(**A**). The correlation between DKK3 expression and IDH. Two-Way ANOVA for multiple comparison test. *** *p* < 0.001 vs. absent DKK3 expression in IDH WT patients; °°° *p* < 0.001 vs. high DKK3 expression in IDH WT patients; ### *p* < 0.001 vs. high DKK3 expression in IDH-mutant patients; §§§ *p* < 0.001 vs. absent DKK3 expression in IDH-mutant patients. (**B**). The correlation between DKK3 expression, IDH, and survival. Two-Way ANOVA for multiple comparison test. Row Factor = 108; F value (5, 33) = 0.450; *p* value = 0.70; Column Factor = 446; F value (7, 33) = 1.85; *p* value = 0.11.

**Figure 8 ijms-25-04091-f008:**
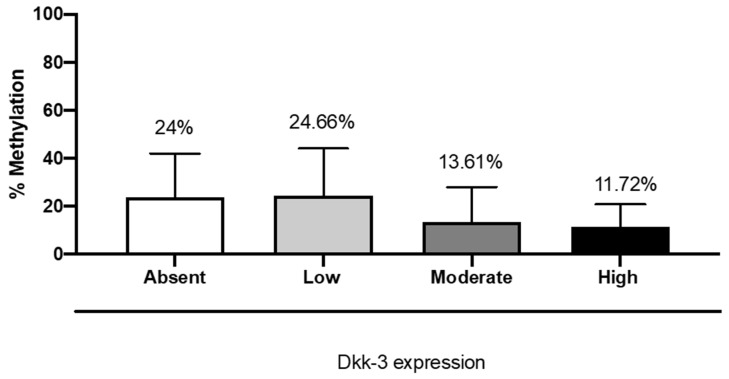
The correlation between DKK3 expression and MGMT methylation status. One-Way ANOVA for multiple comparison test. F value = 1.90; *p* value = 0.15; *p* value summary = ns; R squared = 0.137.

**Figure 9 ijms-25-04091-f009:**
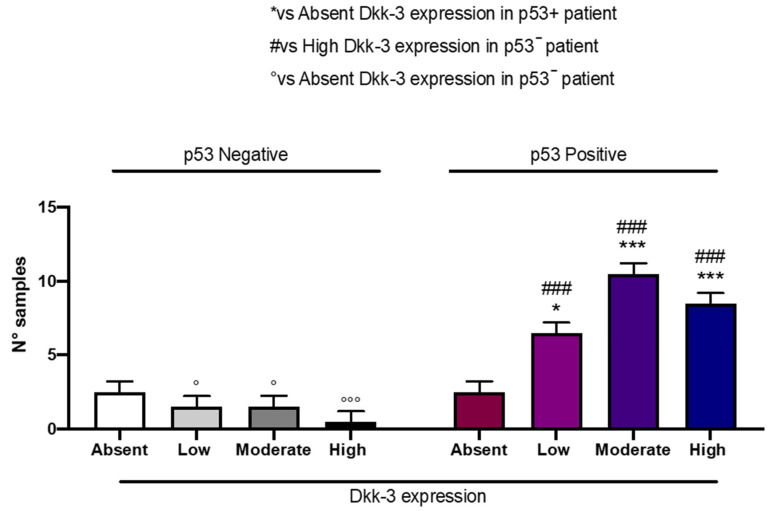
The correlation between DKK3 expression and p53 expression. Two-Way ANOVA for multiple comparison test. * *p* < 0.05 and *** *p* < 0.001 vs. absent DKK3 in p53-Positive patients; ### *p* < 0.001 vs. high DKK3 expression in p53-Negative patients; ° *p* < 0.05 and °°° *p* < 0.001 vs. absent DKK3 expression in p53-Negative patients.

**Figure 10 ijms-25-04091-f010:**
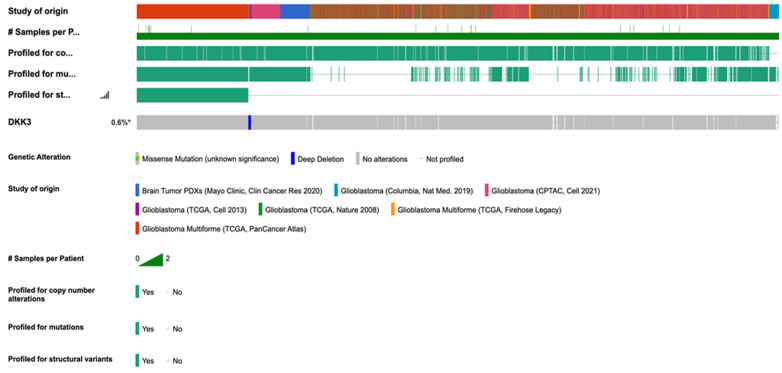
Combined study (2241 samples). Querying 2241 samples/2220 patients in 7 studies of DKK3. #: samples for patients; *: % of DKK3.

**Figure 11 ijms-25-04091-f011:**
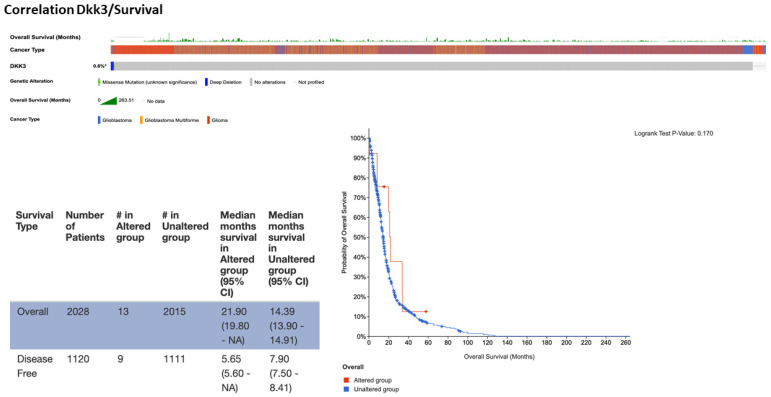
Correlation of DKK3/survival. The analysis performed on 2028 patients showed an alteration of DKK3 in only 13 patients, compared to 2015 patients who presented unaltered DKK3. Interestingly, the median months of survival in the altered DKK3 group was higher (21.9 months) compared to 14.9 months of survival observed in patients with unaltered DKK3, suggesting a protective effect of DKK3 on patient survival. #: samples for patients; *: % of DKK3.

**Table 1 ijms-25-04091-t001:** Patient data.

	Clinical Patient Characteristics	Cases
Sex	Women	24
	Men	16
Tumor Site	Frontal	15
	Temporal	10
	Fronto-Insular	6
	Temporo-Parietal	6
	Multilobar	3
Symptoms	Headache	40
	Seizures	14
	Personality Changes	10
	Increased Intracranial Pressure	10
Neurological Findings	Hemiparesis	23
	Central-Type Facial Nerve Palsy	14
	Babinski Sign	10
	Papilledema	8
	Unremarkable	6
Surgery	Biopsy	2
	Craniotomy: Maximum Resection	12
	Craniotomy: Supratotal Resection	26
Adjuvant Therapy	Stupp Protocol	39

**Table 2 ijms-25-04091-t002:** Biomolecular data and survival for each patient.

Patient	DKK3	Ki67 (%)	IDH1	MGMT Promoter Methylation (%)	P53	Survival (Months)
1	2	40	WT	6	+	9
2	0	20	WT	8	-	2
3	2	30	WT	6	+	47
4	0	40	WT	20	+	11
5	0	80	WT	8	+	1
6	2	40	WT	6	+	7
7	2	10	WT	8	-	1
8	2	50	WT	8	-	3
9	2	10	R132H	45	+	54
10	2	80	R132C	45	+	23
11	1	>20	WT	6	-	15
12	2	>50	WT	30	+	1
13	1	>20	WT	55	+	22
14	2	>50	WT	6	+	19
15	1	20	WT	55	-	44
16	2	25	WT	7	+	4
17	2	20	WT	4	+	13
18	0	10	WT	27	-	28
19	1	>35	R132H	4	+	19
20	2	>50	WT	4	+	14
21	1	10	WT	30	+	8
22	1	20	WT	25	+	1
23	2	15	WT	10	+	43
24	0	50	WT	40	+	10
25	2	15	WT	8	+	1
26	2	10	WT	6	+	6
27	1	20	R132H	25	+	2
28	2	10	R132H	10	+	2
29	2	15	WT	6	+	50
30	2	10	WT	5	+	21
31	1	30	WT	12	+	16
32	2	10	WT	20	-	3
33	2	15	R132X	8	+	20
34	2	10	R132H	10	+	21
35	2	10	WT	10	+	43
36	2	20	WT	8	+	2
37	2	10	WT	10	+	11
38	1	30	WT	30	+	27
39	0	80	WT	55	-	18
40	2	20	R132H	10	+	13

## Data Availability

The data presented in this study are available on request from the corresponding author.

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
