# Peer review of "DKK3 Expression in Glioblastoma: Correlations with Biomolecular Markers"

_ijms, 2024, doi:10.3390/ijms25074091_

Round 1

Reviewer 1 Report

Comments and Suggestions for Authors

A. Major concerns

1. The patient cohort, 40 patients (32 IDH1/2 wild-type and 8 IDH1 mutant) is too small to support the many conclusions made by the authors. In many instances, the word "significant" is not supported by their data. Numerous specific examples are listed in the next section labelled as "Specific Concerns".

2. Given that only 13 patients out of 2220 glioblastoma patients harboured alterations in Dkk3 (just over 0.5%), the results from the cBIOPORTAL database themselves do not validate the patient cohort derived data from immunohistochemistry (IHC) and Western blotting.

3. A table should be included that summarizes Lines 171-206, such as summarizing the clinical patient characteristics, including demographics (age, sex, survival, etc.). Other than these details, the authors are presenting standard clinical care matters (clinical presentation, neuroimaging, neurosurgery) with which the readers will be familiar.

4. The discussion is very long. This is not a review article; it is a research based manuscript, so lengthy explanations of all of the lab based parameters are not necessary; everything can be briefer in this regard.

5. The authors could assess the expression of activated caspase-3 and compare to Dkk3 expression in an adjacent 4 or 5 micron FFPE section; in fact, there are methods using dual immunofluorescence that could co-label Dkk3 and activated caspase-3.

B. Specific concerns

1. Abstract

a. The authors should mention that they have already functionally validated this hypothesis in vitro and in vivo and provide more details (number of cell lines, gain- and/or loss function of Dkk3, model systems, etc. in the revised Introduction.

b. Lines 22 and 67-68: MGMT promoter methylation and IDH status are not "remarkable" biomolecular findings; they are standard assessments in the contemporary neuropathology/molecular pathology clinical laboratory at tertiary care medical centres.

2. Introduction

a. As stated above, provide more details about the in vitro and in vivo modelling of your hypothesis as published in reference #18.

3. Materials and Methods

a. Not all samples included in your analysis from the cBioPortal have been evaluated using the most version of the WHO CNS (2021) criteria. Please clarify. Was a neuropathologist part of the study team?

4. Results

a. Figure 1: Add higher power insets to better visualize the tumour endothelial cells and also add scale bars.

b. Figures 2, 3, 4, 5, 6: The interpretations and conclusions of the data (correlations with sex, age, Ki67, ), despite the applied statistics, are based on very small sample numbers and as such, are far too preliminary to support the authors' conclusions. At worst, the authors frequently over-interpret the data. At best, the authors have observed trends that must be validated in a much larger patient cohort utilizing actual patient samples rather than extracting from publicly available datasets.

c. Section 3.4: For lines 247-251, Ki67 (or MIB1) labelling has been available for over 30 years; the authors findings are not novel in this regard. Hence, it is unclear how individual cases can be interpreted as the authors have done.

d. Section 3.5: Based on the authors' own data, there is absolutely no correlation between Dkk3 expression and IDH status. The clinical and biological significance of their data, as demonstrated in Figure 7 is unclear. Moreover, the number of patients with IDH mutations (8) is too low to support the authors' conclusions. As such, the authors need to show considerable caution in how they interpret this data.

e. Section 3.8 Correlations with cBioPortal

Lines 299-301: The statement "highlighting the protective value of Dkk3 in postponing disease onset" is not supported by the data and should be deleted or significantly restated.

Line 310: Primary GBM (WHO 2021) excludes those tumours with evidence of any IDH mutation.

Lines 318-319: The statement "suggesting the protective role of Dkk3 in promoting MGMT methylation" is not supported by the data presented; this is speculative at best and probably should be deleted or completely restated.

Figure 11: In fact, the KM survival curves show NS (no significance) between altered and unaltered Dkk3 status.

5. Discussion

Lines 352-353: Positive immunostaining of Dkk3 in tumour endothelial cells does not suggest that Dkk3 is linked to tumour neoangiogenesis. This is highly speculative.

Lines 354-356: Similarly, reduced expression of Dkk3 in tumour cells is consistent with its role as a tumour suppressor but does not validate that its downregulation can play a role in glioma progression, "mediated by a dysregulation of apoptosis". This remains as a hypothesis to be tested.

Lines 359-363: Although, from the small cohort, there is a trend to see increased survival in glioblastomas with higher levels of Dkk3 expression, this was not confirmed or validated using the cBioPOrtal; the latter results were not statistically significant and hence do not support the observations in the clinical patient cohort.

Lines 366-382: This entire paragraph on sex differences and clinical outcomes in glioblastoma is entirely speculative and is not supported by the relatively small patient numbers in this pilot study. At best, this is a hypothesis that can be tested prospectively. I recommend significant reduction of this portion of the Discussion.

Lines 383-392: Similarly, the paragraph devoted to age, glioblastoma and Dkk3 expression represents an over-interpretation of the data which is also not validated by the cBioPortal data.

Lines 403-409: It is unclear why the authors completely restate the results of the Ki67 marker studies in the Discussion.

Lines 409-412: This represents another example of over-interpretation of the data. Furthermore, apoptosis wasn't assessed in the clinical patient cohort so this portion of the Discussion is also speculative.

Lines 430-433: The authors have not correctly represented their data on IDH status and Dkk3 levels since there was no correlation between these markers.

Lines 437-440: Rather than consider high(er) levels of Dkk3 expression and IDH mutant status as linked to better patient prognosis, their data, in fact, support that these markers are uncoupled and probably independent variables.

Lines 440-442: Whilst we can agree to link Dkk3 to the Wnt pathway, it is unclear that including IDH mutant status together is correct based upon the data presented in the manuscript.

Lines 455-461: It is not necessary to completely restate the MGMT promoter methylation data in the Discussion. Summarize the key lessons learned.

Lines 464-466: Again, the results from cBioPortal analysis do not link or "confirm" Dkk3 expression and MGMT promoter methylation.

Lines 479-484: The authors also over-interpret the p53 results and prematurely suggest a role for promoting apoptosis or improving treatment sensitivity to temozolomide.

C. Minor concerns

1. Abstract:

Line 15: Add "survival" after "median".

Line 24: Replace "than" with "and" and add "was" after "analyses".

2. Introduction

Line 49: Add "the" before "Dkk3".

3. Materials and Methods

Table 1:

- For the column labels, clarify IDH1 and MGMT promoter methylation.

- Several patients may have been incorrectly listed with R123H mutatiuons when they had R132H mutations (patients 9, 19, 27 and 28).

Line 93: Add 'a" prior to "dako".

Line 99: Is Mayer's "haemalum" similar to hematoxylin? Please clarify.

Line 113: What is meant by "obtained lysate cells"?

4. Results

Line 179: Replace "seems" with "appears".

Line 185: Add "the" prior to "biopsy".

Line 193: Replace "out of the" with "apart from".

Line 202: Replace "II" with "two".

Line 203: Add 'a" prior to "nitrosourea" (note the correct spelling).

Lines 238 and 245: Change "media" to "median".

Line 245: Replace "resulted to be lowered" with "was lower".

Line 274: Add "the" prior to "MGMT".

Lines 317 and 318: Add "the" prior to "Dkk3" and correct the spelling of "revealed".

5. Discussion

Line 351: The cited reference (#25) is incorrect. Please cite the correct paper.

Line 357: Add "an" prior to "inhibitor".

Line 361: Change "media" to "median".

Line 429: Replace "about" with "the".

Line 453: Replace "showed" with "correlated".

Line 474: Replace "and" with "an".

Lines 476-477: Delete "A". Start the next line with "Recent".

Line 479: Add "the" before "WNT".

Line 493: Rewrite as: "permit better stratification of GB patients".

Comments on the Quality of English Language

This manuscript needs to be read carefully by a primary English language speaker, preferably a clinician with an understanding of cancer.

Author Response

RE: “Dkk3 Expression in Glioblastoma: Correlations with Biomolecular Markers. Preliminary Results

Manuscript ID: ijms-2907354

Response to your comments

Reviewer #1

We thank Reviewer for suggestions. We have carefully read reviewer’s comments, and have revised our manuscript accordingly.

  1. Major concerns
  2. The patient cohort, 40 patients (32 IDH1/2 wild-type and 8 IDH1 mutant) is too small to support the many conclusions made by the authors. In many instances, the word "significant" is not supported by their data. Numerous specific examples are listed in the next section labelled as "Specific Concerns".
  • We agree with the reviewer's observation regarding our case study. Forty patients certainly do not represent a large number of cases. Our goal was to make known the data we obtained, such that it represents a potential starting point for further research with a larger number of patients. As proof of this, in the title of the manuscript, we write "Preliminary Results", and, in Discussion section, we state that "The results of our study can lay the underpinnings for carrying studies on a large number of patients to confirm role of Dkk3 in GB initiation and progression". In any case, in accordance with the reviewer's indications, we have modified the text, also removing terms and statements that could have created inaccurate interpretations.

  1. Given that only 13 patients out of 2220 glioblastoma patients harboured alterations in Dkk3 (just over 0.5%), the results from the cBIOPORTAL database themselves do not validate the patient cohort derived data from immunohistochemistry (IHC) and Western blotting.
  • We thank the reviewer’s comment, we can better explain our hypothesis and revise our manuscript. In consideration of the low number of publications concerning the role of the WNT pathway and more specifically the role of Dkk3 in glioblastomas, we wanted to correlate and compare our data by turning to the cBioportal database, which includes studies with a greater number of cases. We therefore found only 5 international studies involving the Dkk3 gene in GB. Therefore, as pointed out by the reviewer, we have changed some terms in the revised text.

  1. A table should be included that summarizes Lines 171-206, such as summarizing the clinical patient characteristics, including demographics (age, sex, survival, etc.). Other than these details, the authors are presenting standard clinical care matters (clinical presentation, neuroimaging, neurosurgery) with which the readers will be familiar.
  • In accordance with your suggestion, we have inserted a new table (Table 2) in the revised manuscript.

Table 2. Data Patients

Cases

Sex

Women

24

Man

16

Tumor Site

Frontal

15

Temporal

10

Fronto-Insular

6

Temporo-Parietal

6

Multilobar

3

Symptoms

Headache

40

Seizures

14

Personality   Changes

10

Increased Intracranial Pressure

10

Neurological Findings

Hemiparesis

23

Central Type Facial Nerve Palsy

14

Babinsky Sign

10

Papilledema

8

Unremarkable

6

Surgery

Biopsy

2

Craniotomy: Maximum Resection

12

Craniotomy: Supratotal Resection

26

Adjuvant Therapy

Stupp Protocol

39

  1. The discussion is very long. This is not a review article; it is a research based manuscript, so lengthy explanations of all of the lab based parameters are not necessary; everything can be briefer in this regard.
  • According to your suggestion, we have shortened the Discussion section.

  1. The authors could assess the expression of activated caspase-3 and compare to Dkk3 expression in an adjacent 4 or 5 micron FFPE section; in fact, there are methods using dual immunofluorescence that could co-label Dkk3 and activated caspase-3.
  • Unfortunately, with great regret, the authors are not able to perform a co-localization for Dkk3 and caspase-3 through immunofluorescence analysis, as it is unable to obtain additional slides from the Lab of Neuropathology to carry out additional analysis. However, in literature, it has been demonstrated the correlation between Dkk3 and caspase-3; particularly, in malignant glioma cells, it is known that Dkk3 induces apoptosis via caspase-3 cleavage by caspase-9, promoting cell type-specific caspase-dependent apoptosis via c-Jun N-terminal kinase (JNK)-dependent signaling activation (Dickkopf-3 in Human Malignant Tumours: A Clinical Viewpoint. Anticancer Research Nov 2020, 40 (11) 5969-5979).

  1. Specific concerns

  1. Abstract
  2. The authors should mention that they have already functionally validated this hypothesis in vitro and in vivo and provide more details (number of cell lines, gain- and/or loss function of Dkk3, model systems, etc. in the revised Introduction.
  • Accordingly, as suggested by reviewer, the authors described in Introduction section, the validated hypothesis in vitro and in vivo as described and confirmed in a previous study (Casili G et al. TLR-4/Wnt modulation as new therapeutic strategy in the treatment of glioblastomas. Oncotarget 2018, 9:37564-37580).

In our previous study we investigated “in vitro” and “in vivo” the biological role of Dkk3 in tumor cells of GB, founding that a modulation of this protein could promote apoptosis, providing a potential strategy in the treatment of GB [18]. Particularly, a significant decreasing of Dkk3 was observed in human glioblastoma cell lines, as well as in U-87 MG xenograft tumors and in GB human patient’s tissues, highlighting that a combined modulation of Wnt/Dkk3 pathways, simultaneously targeting apoptosis and survival signaling defects, might shift the balance from tumor growth stasis to cytotoxic therapeutic responses, flowing in greater therapeutic benefits.

  1. Lines 22 and 67-68: MGMT promoter methylation and IDH status are not "remarkable" biomolecular findings; they are standard assessments in the contemporary neuropathology/molecular pathology clinical laboratory at tertiary care medical centres.
  • As suggested by the reviewer, we have changed the sentences in the revised text.

In addition, we assessed with statistical analysis, correlations between expression of Dkk3 and overall survival, age, gender, Ki-67, p53, MGMT and IDH status.

  1. Introduction
  2. As stated above, provide more details about the in vitro and in vivo modelling of your hypothesis as published in reference #18.
  • Accordingly, as suggested by reviewer, the authors described in Introduction section, the validated hypothesis related to Dkk3 confirmed in a previous study (Casili G et al. TLR-4/Wnt modulation as new therapeutic strategy in the treatment of glioblastomas. Oncotarget 2018, 9:37564-37580).

In our previous study we investigated “in vitro” and “in vivo” the biological role of Dkk3 in tumor cells of GB, founding that a modulation of this protein could promote apoptosis, providing a potential strategy in the treatment of GB [18]. Particularly, a significant decreasing of Dkk3 was observed in human glioblastoma cell lines, as well as in U-87 MG xenograft tumors and in GB human patient’s tissues, highlighting that a combined modulation of Wnt/Dkk3 pathways, simultaneously targeting apoptosis and survival signaling defects, might shift the balance from tumor growth stasis to cytotoxic therapeutic responses, flowing in greater therapeutic benefits.

  1. Materials and Methods
  2. Not all samples included in your analysis from the cBioPortal have been evaluated using the most version of the WHO CNS (2021) criteria. Please clarify. Was a neuropathologist part of the study team?
  • The authors better specified that a combined study was performed (2241 samples), in which 2241 samples / 2220 patients were queried through cBioPortal. Specifically, combined study involved different projects also performed prior to 2021 (Glioblastoma, TGCA Cell 2013; Glioblastoma, Columbia Nat. Med. 2019; Glioblastoma, TGCA Nature 2008; Brain Tumors, Mayo Clinic, Clinic Cancer Res 2020).

  1. Results
  2. Figure 1: Add higher power insets to better visualize the tumour endothelial cells and also add scale bars.
  • In accordance with the reviewer's suggestions, we have modified Figure 1 in the revised text.

  1. Figures 2, 3, 4, 5, 6: The interpretations and conclusions of the data (correlations with sex, age, Ki67), despite the applied statistics, are based on very small sample numbers and as such, are far too preliminary to support the authors' conclusions. At worst, the authors frequently over-interpret the data. At best, the authors have observed trends that must be validated in a much larger patient cohort utilizing actual patient samples rather than extracting from publicly available datasets.
  • It seems clear that our case history is not numerically relevant, just as the data we obtained and validated by statistical analysis are not conclusive. On the other hand, these data can represent a starting point for further research with a larger number of patients. To confirm this, in the title we write "Preliminary results" and in the discussion section of the text we state "The results of our study can lay the underpinnings for carrying studies on a large number of patients to confirm role of Dkk3 in GB initiation and progression ".

  1. Section 3.4: For lines 247-251, Ki67 (or MIB1) labelling has been available for over 30 years; the authors findings are not novel in this regard. Hence, it is unclear how individual cases can be interpreted as the authors have done.
  • In our study, we correlated the expression of Dkk3 and Ki-67. For the first time, we documented that patients with moderate and/or high Dkk3 expression had reduced Ki-67 expression. Therefore, the relationship we highlighted between Dkk3 expression and cell proliferation could represent, if further confirmed, a possible independent prognostic marker.

  1. Section 3.5: Based on the authors' own data, there is absolutely no correlation between Dkk3 expression and IDH status. The clinical and biological significance of their data, as demonstrated in Figure 7 is unclear. Moreover, the number of patients with IDH mutations (8) is too low to support the authors' conclusions. As such, the authors need to show considerable caution in how they interpret this data.
  • We certainly agree with the reviewer that the number of patients in our dataset with IDH mutations is low. However, in a dataset of 40 patients, having 8 cases of grade IV astrocytoma IDH mutated is in line with what has recently been reported in the literature. (Ostrom QT et al. CBTRUS Statistical Report: primary brain and other central nervous system tumors diagnosed in the United States in 2014-2018. Neuro Oncol. 2021; 23(12 Suppl 2):iii1–iii105). Therefore, we reported the data obtained highlighting that "an increased survival, expressed in months, was observed in patients with moderate and high Dkk3 expression, both with IDH wild type and mutated". Furthermore, to confirm the correct observations of the reviewer we also state that "Although these data are not statistically significant, a higher expression of Dkk3, associated with a mutated IDH status, could lead an increase in survival".

  1. Section 3.8 Correlations with cBioPortal

Lines 299-301: The statement "highlighting the protective value of Dkk3 in postponing disease onset" is not supported by the data and should be deleted or significantly restated.

  • In accordance with the reviewer's observation, in Results Section, Subsection 3.8. Correlations with cBioportal Database, we modified the sentence: "in patients with Dkk3 mutated (altered group), the incidence age ranged from 25 until 80 years, hypothesizing an involvement of Dkk3 in postponing disease onset”.

Line 310: Primary GBM (WHO 2021) excludes those tumours with evidence of any IDH mutation.

  • In accordance with the reviewer's observation we have modified the text and removed the sentence. In addition we also modified the text, in line of the most recent classification, both in Introduction and in Material and Methods.

Lines 318-319: The statement "suggesting the protective role of Dkk3 in promoting MGMT methylation" is not supported by the data presented; this is speculative at best and probably should be deleted or completely restated.

  • Accordingly, as suggested by reviewer, the authors deleted the statement above mentioned.

Figure 11: In fact, the KM survival curves show NS (no significance) between altered and unaltered Dkk3 status.

  • Accordingly, as suggested by reviewer, the authors deleted the statement above mentioned.

  1. Discussion

Lines 352-353: Positive immunostaining of Dkk3 in tumour endothelial cells does not suggest that Dkk3 is linked to tumour neoangiogenesis. This is highly speculative.

  • According to the reviewer's observations, in the revised text, we have removed the sentence.

Lines 354-356: Similarly, reduced expression of Dkk3 in tumour cells is consistent with its role as a tumour suppressor but does not validate that its downregulation can play a role in glioma progression, "mediated by a dysregulation of apoptosis". This remains as a hypothesis to be tested. 

  • In agreement with the reviewer's observations, in the revised text, we modified the sentence: "These data confirm our previous results, suggesting a role on tumor growth of Dkk3, due to its loss of function as tumor-suppression-gene [18]. In addition, as the reviewer suggested in a previous comment, we also added in Introduction that in our previous study we investigated “in vitro” and “in vivo” the biological role of Dkk3 in tumor cells of GB, founding that a modulation of this protein could promote apoptosis, providing a potential strategy in the treatment of GB [18]. Particularly, a significant decreasing of Dkk3 was observed in human glioblastoma cell lines, as well as in U-87 MG xenograft tumors and in GB human patient’s tissues, highlighting that a combined modulation of Wnt/Dkk3 pathways, simultaneously targeting apoptosis and survival signaling defects, might shift the balance from tumor growth stasis to cytotoxic therapeutic responses, flowing in greater therapeutic benefits.

Lines 359-363: Although, from the small cohort, there is a trend to see increased survival in glioblastomas with higher levels of Dkk3 expression, this was not confirmed or validated using the cBioPortal; the latter results were not statistically significant and hence do not support the observations in the clinical patient cohort.

  • Accordingly, as suggested by reviewer, the authors deleted the sentence above mentioned in the revised text.

Lines 366-382: This entire paragraph on sex differences and clinical outcomes in glioblastoma is entirely speculative and is not supported by the relatively small patient numbers in this pilot study. At best, this is a hypothesis that can be tested prospectively. I recommend significant reduction of this portion of the Discussion.

  • In accordance with the reviewer's observations, in the revised text, we have reduced the part of the discussion regarding sex difference and clinical outcomes in glioblastoma.

Lines 383-392: Similarly, the paragraph devoted to age, glioblastoma and Dkk3 expression represents an over-interpretation of the data which is also not validated by the cBioPortal data.

  • As suggested by the reviewer, we have rewritten this paragraph.

Lines 403-409: It is unclear why the authors completely restate the results of the Ki67 marker studies in the Discussion.

  • As suggested by the reviewer, we have reduced, in the revised text, the paragraph dedicated to the Ki-67.

Lines 409-412: This represents another example of over-interpretation of the data. Furthermore, apoptosis wasn't assessed in the clinical patient cohort so this portion of the Discussion is also speculative.

  • As suggested by the reviewer, this part of the discussion section, in the revised text, has been shortened and rewritten.

Lines 430-433: The authors have not correctly represented their data on IDH status and Dkk3 levels since there was no correlation between these markers.

  • In this study, we have reported the data obtained and, although, as already written in the text, they are not statistically significant. However, they are certainly interesting as they can represent the starting point for further and broader research. We have, however, better rewritten this part in the revised text, in accordance with the reviewer's observations.

Major advances in cancer genetics have revealed that the genes encoding IDHs are frequently mutated in a variety of human malignancies, including GB [30-31]. More than 90% of the mutations in IDH1 are R132H that has been associated with significantly improved prognosis and longer progression-free and overall survival [32-33]. IDH1-R132H mutation leads to both a less aggressive phenotype and radiosensitization of glioma cells [34]. For the first time, in this study, we demonstrated that both patients with IDH wild-type GB and IDH mutated grade IV astrocytoma showed a moderate and high expression of Dkk3, compared to the remaining patients (6) with absent or low Dkk3 expression. In addition, to better understand the impact of IDH status (wild type or mutated) on survival, related to Dkk3 expression, we performed a correlation among these three parameters. Interestingly, an in-creased overall survival, was observed in patients with moderate and high Dkk3 expression, both with IDH wild-type and mutated (Fig 7). These data, although interesting, are not statistically significant and, therefore, new and further confirmations are necessary.

Lines 437-440: Rather than consider high(er) levels of Dkk3 expression and IDH mutant status as linked to better patient prognosis, their data, in fact, support that these markers are uncoupled and probably independent variables.

  • In agreement with the reviewer we have modified this part of the text.

Major advances in cancer genetics have revealed that the genes encoding IDHs are frequently mutated in a variety of human malignancies, including GB [30-31]. More than 90% of the mutations in IDH1 are R132H that has been associated with significantly im-proved prognosis and longer progression-free and overall survival [32-33]. IDH1-R132H mutation leads to both a less aggressive phenotype and radiosensitization of glioma cells [34]. For the first time, in this study, we demonstrated that both patients with IDH wild-type GB and IDH mutated grade IV astrocytoma showed a moderate and high expression of Dkk3, compared to the remaining patients (6) with absent or low Dkk3 expression. In addition, to better understand the impact of IDH status (wild type or mutated) on survival, related to Dkk3 expression, we performed a correlation among these three parameters. Interestingly, an increased overall survival, was observed in patients with moderate and high Dkk3 expression, both with IDH wild-type and mutated (Fig 7). These data, although interesting, are not statistically significant and, therefore, new and further confirmations are necessary.

Lines 440-442: Whilst we can agree to link Dkk3 to the Wnt pathway, it is unclear that including IDH mutant status together is correct based upon the data presented in the manuscript.

  • In accordance with the reviewer's observations, we have better rewritten this part in the revised text.

Major advances in cancer genetics have revealed that the genes encoding IDHs are frequently mutated in a variety of human malignancies, including GB [30-31]. More than 90% of the mutations in IDH1 are R132H that has been associated with significantly improved prognosis and longer progression-free and overall survival [32-33]. IDH1-R132H mutation leads to both a less aggressive phenotype and radiosensitization of glioma cells [34]. For the first time, in this study, we demonstrated that both patients with IDH wild-type GB and IDH mutated grade IV astrocytoma showed a moderate and high expression of Dkk3, compared to the remaining patients (6) with absent or low Dkk3 expression. In addition, to better understand the impact of IDH status (wild type or mutated) on survival, related to Dkk3 expression, we performed a correlation among these three parameters. Interestingly, an increased overall survival, was observed in patients with moderate and high Dkk3 expression, both with IDH wild-type and mutated (Fig 7). These data, although interesting, are not statistically significant and, therefore, new and further confirmations are necessary.

Lines 455-461: It is not necessary to completely restate the MGMT promoter methylation data in the Discussion. Summarize the key lessons learned.

  • As suggested by the reviewer, we, in the revised text, have better rewritten.

It has been recently reported that inhibition of WNT pathway downregulates MGMT ex-pression and reestablishes chemosensitivity of DNA-alkylating drugs in mouse models [17]. Interestingly, in our study, patients with moderate and high expression of Dkk3 showed a reduction in % methylation of MGMT gene, compared to patients with absent or low Dkk3 expression, also revealing that cases showing a higher percentage of methylation had a survival of 10 months. Therefore, for the first time, we showed that correlation between Dkk3 and MGMT could have a prognostic role.

Lines 464-466: Again, the results from cBioPortal analysis do not link or "confirm" Dkk3 expression and MGMT promoter methylation.

  • As suggested by the reviewer, we, in the revised text, have better rewritten the text.

It has been recently reported that inhibition of WNT pathway downregulates MGMT ex-pression and reestablishes chemosensitivity of DNA-alkylating drugs in mouse models [17]. Interestingly, in our study, patients with moderate and high expression of Dkk3 showed a reduction in % methylation of MGMT gene, compared to patients with absent or low Dkk3 expression, also revealing that cases showing a higher percentage of methylation had a survival of 10 months. Therefore, for the first time, we showed that correlation between Dkk3 and MGMT could have a prognostic role.

Lines 479-484: The authors also over-interpret the p53 results and prematurely suggest a role for promoting apoptosis or improving treatment sensitivity to temozolomide.

  • As suggested by the reviewer, in the revised text, we, have better rewritten.

p53 plays a central role in maintaining cellular homeostasis and is frequently deregulated in cancer [35]. For the first time, in our study, we demonstrated that GB patients with moderate and high expression of Dkk3 correlated with a significant elevated detectable p53 expression compared to patients with absent Dkk3 expression, confirming our hypothesis that Dkk3 can be considered as a prognostic factor in GB.

  1. Minor concerns

  • Abstract: We, in the revised text, have made the corrections indicated by the reviewer.
  • Introduction: We, in the revised text, have made the corrections indicated by the reviewer.
  • Materials and Methods: We, in the revised text, have made the corrections indicated by the reviewer.

Line 99: Is Mayer's "haemalum" similar to hematoxylin? Please clarify.

Mayer’s haemalum solution is used in H&E as the most widely used histological staining technique. Particularly, nuclear staining is achieved using ready-to-use staining solutions, such as Mayer’s hemalum solution or Hematoxylin solution modified.

Line 113: What is meant by "obtained lysate cells"?

The authors corrected the sentence, highlighting that the meaning was the total cellular proteins obtained in lysates.

  • Results: We, in the revised text, have made the corrections indicated by the reviewer.

RE: “Dkk3 Expression in Glioblastoma: Correlations with Biomolecular Markers. Preliminary Results

Manuscript ID: ijms-2907354

Response to your comments

Reviewer #1

We thank Reviewer for suggestions. We have carefully read reviewer’s comments, and have revised our manuscript accordingly.

  1. Major concerns
  2. The patient cohort, 40 patients (32 IDH1/2 wild-type and 8 IDH1 mutant) is too small to support the many conclusions made by the authors. In many instances, the word "significant" is not supported by their data. Numerous specific examples are listed in the next section labelled as "Specific Concerns".
  • We agree with the reviewer's observation regarding our case study. Forty patients certainly do not represent a large number of cases. Our goal was to make known the data we obtained, such that it represents a potential starting point for further research with a larger number of patients. As proof of this, in the title of the manuscript, we write "Preliminary Results", and, in Discussion section, we state that "The results of our study can lay the underpinnings for carrying studies on a large number of patients to confirm role of Dkk3 in GB initiation and progression". In any case, in accordance with the reviewer's indications, we have modified the text, also removing terms and statements that could have created inaccurate interpretations.

  1. Given that only 13 patients out of 2220 glioblastoma patients harboured alterations in Dkk3 (just over 0.5%), the results from the cBIOPORTAL database themselves do not validate the patient cohort derived data from immunohistochemistry (IHC) and Western blotting.
  • We thank the reviewer’s comment, we can better explain our hypothesis and revise our manuscript. In consideration of the low number of publications concerning the role of the WNT pathway and more specifically the role of Dkk3 in glioblastomas, we wanted to correlate and compare our data by turning to the cBioportal database, which includes studies with a greater number of cases. We therefore found only 5 international studies involving the Dkk3 gene in GB. Therefore, as pointed out by the reviewer, we have changed some terms in the revised text.

  1. A table should be included that summarizes Lines 171-206, such as summarizing the clinical patient characteristics, including demographics (age, sex, survival, etc.). Other than these details, the authors are presenting standard clinical care matters (clinical presentation, neuroimaging, neurosurgery) with which the readers will be familiar.
  • In accordance with your suggestion, we have inserted a new table (Table 2) in the revised manuscript.

Table 2. Data Patients

Cases

Sex

Women

24

Man

16

Tumor Site

Frontal

15

Temporal

10

Fronto-Insular

6

Temporo-Parietal

6

Multilobar

3

Symptoms

Headache

40

Seizures

14

Personality   Changes

10

Increased Intracranial Pressure

10

Neurological Findings

Hemiparesis

23

Central Type Facial Nerve Palsy

14

Babinsky Sign

10

Papilledema

8

Unremarkable

6

Surgery

Biopsy

2

Craniotomy: Maximum Resection

12

Craniotomy: Supratotal Resection

26

Adjuvant Therapy

Stupp Protocol

39

  1. The discussion is very long. This is not a review article; it is a research based manuscript, so lengthy explanations of all of the lab based parameters are not necessary; everything can be briefer in this regard.
  • According to your suggestion, we have shortened the Discussion section.

  1. The authors could assess the expression of activated caspase-3 and compare to Dkk3 expression in an adjacent 4 or 5 micron FFPE section; in fact, there are methods using dual immunofluorescence that could co-label Dkk3 and activated caspase-3.
  • Unfortunately, with great regret, the authors are not able to perform a co-localization for Dkk3 and caspase-3 through immunofluorescence analysis, as it is unable to obtain additional slides from the Lab of Neuropathology to carry out additional analysis. However, in literature, it has been demonstrated the correlation between Dkk3 and caspase-3; particularly, in malignant glioma cells, it is known that Dkk3 induces apoptosis via caspase-3 cleavage by caspase-9, promoting cell type-specific caspase-dependent apoptosis via c-Jun N-terminal kinase (JNK)-dependent signaling activation (Dickkopf-3 in Human Malignant Tumours: A Clinical Viewpoint. Anticancer Research Nov 2020, 40 (11) 5969-5979).

  1. Specific concerns

  1. Abstract
  2. The authors should mention that they have already functionally validated this hypothesis in vitro and in vivo and provide more details (number of cell lines, gain- and/or loss function of Dkk3, model systems, etc. in the revised Introduction.
  • Accordingly, as suggested by reviewer, the authors described in Introduction section, the validated hypothesis in vitro and in vivo as described and confirmed in a previous study (Casili G et al. TLR-4/Wnt modulation as new therapeutic strategy in the treatment of glioblastomas. Oncotarget 2018, 9:37564-37580).

In our previous study we investigated “in vitro” and “in vivo” the biological role of Dkk3 in tumor cells of GB, founding that a modulation of this protein could promote apoptosis, providing a potential strategy in the treatment of GB [18]. Particularly, a significant decreasing of Dkk3 was observed in human glioblastoma cell lines, as well as in U-87 MG xenograft tumors and in GB human patient’s tissues, highlighting that a combined modulation of Wnt/Dkk3 pathways, simultaneously targeting apoptosis and survival signaling defects, might shift the balance from tumor growth stasis to cytotoxic therapeutic responses, flowing in greater therapeutic benefits.

  1. Lines 22 and 67-68: MGMT promoter methylation and IDH status are not "remarkable" biomolecular findings; they are standard assessments in the contemporary neuropathology/molecular pathology clinical laboratory at tertiary care medical centres.
  • As suggested by the reviewer, we have changed the sentences in the revised text.

In addition, we assessed with statistical analysis, correlations between expression of Dkk3 and overall survival, age, gender, Ki-67, p53, MGMT and IDH status.

  1. Introduction
  2. As stated above, provide more details about the in vitro and in vivo modelling of your hypothesis as published in reference #18.
  • Accordingly, as suggested by reviewer, the authors described in Introduction section, the validated hypothesis related to Dkk3 confirmed in a previous study (Casili G et al. TLR-4/Wnt modulation as new therapeutic strategy in the treatment of glioblastomas. Oncotarget 2018, 9:37564-37580).

In our previous study we investigated “in vitro” and “in vivo” the biological role of Dkk3 in tumor cells of GB, founding that a modulation of this protein could promote apoptosis, providing a potential strategy in the treatment of GB [18]. Particularly, a significant decreasing of Dkk3 was observed in human glioblastoma cell lines, as well as in U-87 MG xenograft tumors and in GB human patient’s tissues, highlighting that a combined modulation of Wnt/Dkk3 pathways, simultaneously targeting apoptosis and survival signaling defects, might shift the balance from tumor growth stasis to cytotoxic therapeutic responses, flowing in greater therapeutic benefits.

  1. Materials and Methods
  2. Not all samples included in your analysis from the cBioPortal have been evaluated using the most version of the WHO CNS (2021) criteria. Please clarify. Was a neuropathologist part of the study team?
  • The authors better specified that a combined study was performed (2241 samples), in which 2241 samples / 2220 patients were queried through cBioPortal. Specifically, combined study involved different projects also performed prior to 2021 (Glioblastoma, TGCA Cell 2013; Glioblastoma, Columbia Nat. Med. 2019; Glioblastoma, TGCA Nature 2008; Brain Tumors, Mayo Clinic, Clinic Cancer Res 2020).

  1. Results
  2. Figure 1: Add higher power insets to better visualize the tumour endothelial cells and also add scale bars.
  • In accordance with the reviewer's suggestions, we have modified Figure 1 in the revised text.

  1. Figures 2, 3, 4, 5, 6: The interpretations and conclusions of the data (correlations with sex, age, Ki67), despite the applied statistics, are based on very small sample numbers and as such, are far too preliminary to support the authors' conclusions. At worst, the authors frequently over-interpret the data. At best, the authors have observed trends that must be validated in a much larger patient cohort utilizing actual patient samples rather than extracting from publicly available datasets.
  • It seems clear that our case history is not numerically relevant, just as the data we obtained and validated by statistical analysis are not conclusive. On the other hand, these data can represent a starting point for further research with a larger number of patients. To confirm this, in the title we write "Preliminary results" and in the discussion section of the text we state "The results of our study can lay the underpinnings for carrying studies on a large number of patients to confirm role of Dkk3 in GB initiation and progression ".

  1. Section 3.4: For lines 247-251, Ki67 (or MIB1) labelling has been available for over 30 years; the authors findings are not novel in this regard. Hence, it is unclear how individual cases can be interpreted as the authors have done.
  • In our study, we correlated the expression of Dkk3 and Ki-67. For the first time, we documented that patients with moderate and/or high Dkk3 expression had reduced Ki-67 expression. Therefore, the relationship we highlighted between Dkk3 expression and cell proliferation could represent, if further confirmed, a possible independent prognostic marker.

  1. Section 3.5: Based on the authors' own data, there is absolutely no correlation between Dkk3 expression and IDH status. The clinical and biological significance of their data, as demonstrated in Figure 7 is unclear. Moreover, the number of patients with IDH mutations (8) is too low to support the authors' conclusions. As such, the authors need to show considerable caution in how they interpret this data.
  • We certainly agree with the reviewer that the number of patients in our dataset with IDH mutations is low. However, in a dataset of 40 patients, having 8 cases of grade IV astrocytoma IDH mutated is in line with what has recently been reported in the literature. (Ostrom QT et al. CBTRUS Statistical Report: primary brain and other central nervous system tumors diagnosed in the United States in 2014-2018. Neuro Oncol. 2021; 23(12 Suppl 2):iii1–iii105). Therefore, we reported the data obtained highlighting that "an increased survival, expressed in months, was observed in patients with moderate and high Dkk3 expression, both with IDH wild type and mutated". Furthermore, to confirm the correct observations of the reviewer we also state that "Although these data are not statistically significant, a higher expression of Dkk3, associated with a mutated IDH status, could lead an increase in survival".

  1. Section 3.8 Correlations with cBioPortal

Lines 299-301: The statement "highlighting the protective value of Dkk3 in postponing disease onset" is not supported by the data and should be deleted or significantly restated.

  • In accordance with the reviewer's observation, in Results Section, Subsection 3.8. Correlations with cBioportal Database, we modified the sentence: "in patients with Dkk3 mutated (altered group), the incidence age ranged from 25 until 80 years, hypothesizing an involvement of Dkk3 in postponing disease onset”.

Line 310: Primary GBM (WHO 2021) excludes those tumours with evidence of any IDH mutation.

  • In accordance with the reviewer's observation we have modified the text and removed the sentence. In addition we also modified the text, in line of the most recent classification, both in Introduction and in Material and Methods.

Lines 318-319: The statement "suggesting the protective role of Dkk3 in promoting MGMT methylation" is not supported by the data presented; this is speculative at best and probably should be deleted or completely restated.

  • Accordingly, as suggested by reviewer, the authors deleted the statement above mentioned.

Figure 11: In fact, the KM survival curves show NS (no significance) between altered and unaltered Dkk3 status.

  • Accordingly, as suggested by reviewer, the authors deleted the statement above mentioned.

  1. Discussion

Lines 352-353: Positive immunostaining of Dkk3 in tumour endothelial cells does not suggest that Dkk3 is linked to tumour neoangiogenesis. This is highly speculative.

  • According to the reviewer's observations, in the revised text, we have removed the sentence.

Lines 354-356: Similarly, reduced expression of Dkk3 in tumour cells is consistent with its role as a tumour suppressor but does not validate that its downregulation can play a role in glioma progression, "mediated by a dysregulation of apoptosis". This remains as a hypothesis to be tested. 

  • In agreement with the reviewer's observations, in the revised text, we modified the sentence: "These data confirm our previous results, suggesting a role on tumor growth of Dkk3, due to its loss of function as tumor-suppression-gene [18]. In addition, as the reviewer suggested in a previous comment, we also added in Introduction that in our previous study we investigated “in vitro” and “in vivo” the biological role of Dkk3 in tumor cells of GB, founding that a modulation of this protein could promote apoptosis, providing a potential strategy in the treatment of GB [18]. Particularly, a significant decreasing of Dkk3 was observed in human glioblastoma cell lines, as well as in U-87 MG xenograft tumors and in GB human patient’s tissues, highlighting that a combined modulation of Wnt/Dkk3 pathways, simultaneously targeting apoptosis and survival signaling defects, might shift the balance from tumor growth stasis to cytotoxic therapeutic responses, flowing in greater therapeutic benefits.

Lines 359-363: Although, from the small cohort, there is a trend to see increased survival in glioblastomas with higher levels of Dkk3 expression, this was not confirmed or validated using the cBioPortal; the latter results were not statistically significant and hence do not support the observations in the clinical patient cohort.

  • Accordingly, as suggested by reviewer, the authors deleted the sentence above mentioned in the revised text.

Lines 366-382: This entire paragraph on sex differences and clinical outcomes in glioblastoma is entirely speculative and is not supported by the relatively small patient numbers in this pilot study. At best, this is a hypothesis that can be tested prospectively. I recommend significant reduction of this portion of the Discussion.

  • In accordance with the reviewer's observations, in the revised text, we have reduced the part of the discussion regarding sex difference and clinical outcomes in glioblastoma.

Lines 383-392: Similarly, the paragraph devoted to age, glioblastoma and Dkk3 expression represents an over-interpretation of the data which is also not validated by the cBioPortal data.

  • As suggested by the reviewer, we have rewritten this paragraph.

Lines 403-409: It is unclear why the authors completely restate the results of the Ki67 marker studies in the Discussion.

  • As suggested by the reviewer, we have reduced, in the revised text, the paragraph dedicated to the Ki-67.

Lines 409-412: This represents another example of over-interpretation of the data. Furthermore, apoptosis wasn't assessed in the clinical patient cohort so this portion of the Discussion is also speculative.

  • As suggested by the reviewer, this part of the discussion section, in the revised text, has been shortened and rewritten.

Lines 430-433: The authors have not correctly represented their data on IDH status and Dkk3 levels since there was no correlation between these markers.

  • In this study, we have reported the data obtained and, although, as already written in the text, they are not statistically significant. However, they are certainly interesting as they can represent the starting point for further and broader research. We have, however, better rewritten this part in the revised text, in accordance with the reviewer's observations.

Major advances in cancer genetics have revealed that the genes encoding IDHs are frequently mutated in a variety of human malignancies, including GB [30-31]. More than 90% of the mutations in IDH1 are R132H that has been associated with significantly improved prognosis and longer progression-free and overall survival [32-33]. IDH1-R132H mutation leads to both a less aggressive phenotype and radiosensitization of glioma cells [34]. For the first time, in this study, we demonstrated that both patients with IDH wild-type GB and IDH mutated grade IV astrocytoma showed a moderate and high expression of Dkk3, compared to the remaining patients (6) with absent or low Dkk3 expression. In addition, to better understand the impact of IDH status (wild type or mutated) on survival, related to Dkk3 expression, we performed a correlation among these three parameters. Interestingly, an in-creased overall survival, was observed in patients with moderate and high Dkk3 expression, both with IDH wild-type and mutated (Fig 7). These data, although interesting, are not statistically significant and, therefore, new and further confirmations are necessary.

Lines 437-440: Rather than consider high(er) levels of Dkk3 expression and IDH mutant status as linked to better patient prognosis, their data, in fact, support that these markers are uncoupled and probably independent variables.

  • In agreement with the reviewer we have modified this part of the text.

Major advances in cancer genetics have revealed that the genes encoding IDHs are frequently mutated in a variety of human malignancies, including GB [30-31]. More than 90% of the mutations in IDH1 are R132H that has been associated with significantly im-proved prognosis and longer progression-free and overall survival [32-33]. IDH1-R132H mutation leads to both a less aggressive phenotype and radiosensitization of glioma cells [34]. For the first time, in this study, we demonstrated that both patients with IDH wild-type GB and IDH mutated grade IV astrocytoma showed a moderate and high expression of Dkk3, compared to the remaining patients (6) with absent or low Dkk3 expression. In addition, to better understand the impact of IDH status (wild type or mutated) on survival, related to Dkk3 expression, we performed a correlation among these three parameters. Interestingly, an increased overall survival, was observed in patients with moderate and high Dkk3 expression, both with IDH wild-type and mutated (Fig 7). These data, although interesting, are not statistically significant and, therefore, new and further confirmations are necessary.

Lines 440-442: Whilst we can agree to link Dkk3 to the Wnt pathway, it is unclear that including IDH mutant status together is correct based upon the data presented in the manuscript.

  • In accordance with the reviewer's observations, we have better rewritten this part in the revised text.

Major advances in cancer genetics have revealed that the genes encoding IDHs are frequently mutated in a variety of human malignancies, including GB [30-31]. More than 90% of the mutations in IDH1 are R132H that has been associated with significantly improved prognosis and longer progression-free and overall survival [32-33]. IDH1-R132H mutation leads to both a less aggressive phenotype and radiosensitization of glioma cells [34]. For the first time, in this study, we demonstrated that both patients with IDH wild-type GB and IDH mutated grade IV astrocytoma showed a moderate and high expression of Dkk3, compared to the remaining patients (6) with absent or low Dkk3 expression. In addition, to better understand the impact of IDH status (wild type or mutated) on survival, related to Dkk3 expression, we performed a correlation among these three parameters. Interestingly, an increased overall survival, was observed in patients with moderate and high Dkk3 expression, both with IDH wild-type and mutated (Fig 7). These data, although interesting, are not statistically significant and, therefore, new and further confirmations are necessary.

Lines 455-461: It is not necessary to completely restate the MGMT promoter methylation data in the Discussion. Summarize the key lessons learned.

  • As suggested by the reviewer, we, in the revised text, have better rewritten.

It has been recently reported that inhibition of WNT pathway downregulates MGMT ex-pression and reestablishes chemosensitivity of DNA-alkylating drugs in mouse models [17]. Interestingly, in our study, patients with moderate and high expression of Dkk3 showed a reduction in % methylation of MGMT gene, compared to patients with absent or low Dkk3 expression, also revealing that cases showing a higher percentage of methylation had a survival of 10 months. Therefore, for the first time, we showed that correlation between Dkk3 and MGMT could have a prognostic role.

Lines 464-466: Again, the results from cBioPortal analysis do not link or "confirm" Dkk3 expression and MGMT promoter methylation.

  • As suggested by the reviewer, we, in the revised text, have better rewritten the text.

It has been recently reported that inhibition of WNT pathway downregulates MGMT ex-pression and reestablishes chemosensitivity of DNA-alkylating drugs in mouse models [17]. Interestingly, in our study, patients with moderate and high expression of Dkk3 showed a reduction in % methylation of MGMT gene, compared to patients with absent or low Dkk3 expression, also revealing that cases showing a higher percentage of methylation had a survival of 10 months. Therefore, for the first time, we showed that correlation between Dkk3 and MGMT could have a prognostic role.

Lines 479-484: The authors also over-interpret the p53 results and prematurely suggest a role for promoting apoptosis or improving treatment sensitivity to temozolomide.

  • As suggested by the reviewer, in the revised text, we, have better rewritten.

p53 plays a central role in maintaining cellular homeostasis and is frequently deregulated in cancer [35]. For the first time, in our study, we demonstrated that GB patients with moderate and high expression of Dkk3 correlated with a significant elevated detectable p53 expression compared to patients with absent Dkk3 expression, confirming our hypothesis that Dkk3 can be considered as a prognostic factor in GB.

  1. Minor concerns

  • Abstract: We, in the revised text, have made the corrections indicated by the reviewer.
  • Introduction: We, in the revised text, have made the corrections indicated by the reviewer.
  • Materials and Methods: We, in the revised text, have made the corrections indicated by the reviewer.

Line 99: Is Mayer's "haemalum" similar to hematoxylin? Please clarify.

Mayer’s haemalum solution is used in H&E as the most widely used histological staining technique. Particularly, nuclear staining is achieved using ready-to-use staining solutions, such as Mayer’s hemalum solution or Hematoxylin solution modified.

Line 113: What is meant by "obtained lysate cells"?

The authors corrected the sentence, highlighting that the meaning was the total cellular proteins obtained in lysates.

  • Results: We, in the revised text, have made the corrections indicated by the reviewer.

Reviewer 2 Report

Comments and Suggestions for Authors

In this article, Caffo et al analyze DKK3 expression by Immunohistochemistry and Western Blot in 40 cases of Glioblastoma, and its correlation with epidemiologic information or other genetic markers. Also, they analyze their results along with data obtained from the cbioportal database. Please find below my comments:

-        English editing is needed

-        Biomolecular analysis, line 135 : the 4th mutation of IDH1, R132S instead of R172S?

-        Why did the authors decide to perform IHC and WB analyses for DKK3, and why not sequencing? Especially that the complete the analyses with data from obtained from the cbioportal where DKK3 mutations were reported.

-        Section 3.1: About ID score. I suggest to add an interpretation for this score, maybe in the Methods section

-        Section 3.2: a significant correlation was observed. However, the numbers are not shown neither in the text, nor in figure 4. P values?

-        Also, in the same section, line 219: LOW expression?  Tor better to say “LOWER” to be more accurate

-        Figure 3: Please add a legend to the Y axis, and unit

-        Figures 5, 6, 7B and 8: Where are the values of the statistic tests performed. This information is missing in the text and figures. The presence or absence of statistical significances should be clearly presented.

-        In my point of view, the discussion part is not clear and not attractive to the readers. It is divided into different segments, and each part talks about a specific change/gene. The discussion needs to be restructured, so it can be clearer and more straightforward.

-        Also, some conclusions and discussions need to be toned down and drawn based on more evidence from the paper.  

Comments on the Quality of English Language

English editing needed. 

Round 2

Reviewer 1 Report

Comments and Suggestions for Authors

The authors have responded to the prior review and made substantial progress in  revising the paper. However, several corrections (in English language) and clarifications remain.

A. Specific concerns

1. Table 1

For 4 patients (#10, 33, 34, 40) you list R312H-, but this wasn't tested for. Please revise and/or explain.

2. Discussion

Lines 373-375 require revision as follows to reflect your findings more accurately: Delete "can be significant to demonstrate" and replace with "support" prior to "a relationship".  Replace "can" with "could" prior to "be used". Add "with further study" after "prognostic markers".

B. Minor concerns

1. Abstract

Line 19: Correct the spelling of "immunohistochemical".

Line 25: Correct the spelling of "exists" and add "the" prior to "literature".

Line 26: Add "the" prior to "WNT".

2. Introduction

Line 49: Delete "the" prior to "Dkk3".

Line 58: Add "expressed" prior to "enzyme".

Line 63: Revise "founding" to "finding".

Line 69: Change "flowing" to "resulting".

Line 74: Add "the" prior to "cBio".

3. Methods

Line 91: Change "mu" (second letter is Greek) to "um" (first letter is Greek)

4. Results

Table 2: Change "Man" to "Men". Correct "Babinsky" to "Babinski".

Line 209: Revise the title of the Table to "Patient Data".

Line 243: Revise "absence" to "absent".

Line 322: Add "the" prior to "Dkk3".

Line 323: Add "the" prior to "Dkk3".

5. Discussion

Lines 348-349: Delete "resulted, both at" and replace with "in" prior to "immunohistochemistry". Replace "than" with "and" prior to "Western-blot". Add "was" prior to "reduced".

Line 365: Change "reaches" to "reaching".

Line 389: Add "to" prior to "an increase".

Line 400: Replace "confirming" with "supporting".

Comments on the Quality of English Language

Several errors in English grammar, syntax and spelling have been identified and listed under "Minor Concerns" in the Response to Authors above.

Author Response

RE: “DKK3 Expression in Glioblastoma: Correlations with Biomolecular Markers.

Manuscript ID: ijms-2907354

Response to your comments

Reviewer #1

We thank Reviewer for suggestions. We have read reviewer’s comments, and have revised our manuscript accordingly.

  1. Specific concerns

  1. Table 1

For 4 patients (#10, 33, 34, 40) you list R312H-, but this wasn't tested for. Please revise and/or explain.

  • We apologize to the reviewer for the errors highlighted in Table 1. We have carefully rechecked Table 1 and, accordingly, have corrected the reported data.

Table 1. Biomolecular data and survival of each patient.

Patient

Dkk3

Ki67 (%)

IDH1

MGMT promoter Methylation (%)

P53

Survival (months)

1

2

40

WT

6

+

9

2

0

20

WT

8

-

2

3

2

30

WT

6

+

47

4

0

40

WT

20

+

11

5

0

80

WT

8

+

1

6

2

40

WT

6

+

7

7

2

10

WT

8

-

1

8

2

50

WT

8

-

3

9

2

10

R132H

45

+

54

10

2

80

R132C

45

+

23

11

1

>20

WT

6

-

15

12

2

>50

WT

30

+

1

13

1

>20

WT

55

+

22

14

2

>50

WT

6

+

19

15

1

20

WT

55

-

44

16

2

25

WT

7

+

4

17

2

20

WT

4

+

13

18

0

10

WT

27

-

28

19

1

>35

R132H

4

+

19

20

2

>50

WT

4

+

14

21

1

10

WT

30

+

8

22

1

20

WT

25

+

1

23

2

15

WT

10

+

43

24

0

50

WT

40

+

10

25

2

15

WT

8

+

1

26

2

10

WT

6

+

6

27

1

20

R132H

25

+

2

28

2

10

R132H

10

+

2

29

2

15

WT

6

+

50

30

2

10

WT

5

+

21

31

1

30

WT

12

+

16

32

2

10

WT

20

-

3

33

2

15

R132X

8

+

20

34

2

10

R132H

10

+

21

35

2

10

WT

10

+

43

36

2

20

WT

8

+

2

37

2

10

WT

10

+

11

38

1

30

WT

30

+

27

39

0

80

WT

55

-

18

40

2

20

R132H

10

+

13

2 Discussion

Lines 373-375 require revision as follows to reflect your findings more accurately: Delete "can be significant to demonstrate" and replace with "support" prior to "a relationship".  Replace "can" with "could" prior to "be used". Add "with further study" after "prognostic markers".

  • In accordance with the reviewer's suggestions, we have modified this part of the Discussion.

Our results support a relationship between Dkk3 expression and cellular proliferation, as well as, their correlation could be used as independent further study.

  1. Minor concerns

  1. Abstract

Line 19: Correct the spelling of "immunohistochemical".

Line 25: Correct the spelling of "exists" and add "the" prior to "literature".

Line 26: Add "the" prior to "WNT".

  • In accordance with the reviewer's suggestions, we have modified the abstract.

Methods: We performed, in a series of 40 patients, immunohistochemical and western-blot evalua-tion of Dkk3 to better understand how the expression of this protein can influence clinical behav-ior, assessing a statistical analysis, with correlations between expression of Dkk3 and overall sur-vival, age, sex, Ki-67, p53, and MGMT and IDH status. We also validate our data correlating with information included in cBioPortal database. Results: In our analyses, Dkk3 expression, both at immunohistochemistry and Western-blot analyses, was reduced or absent in many cases showing a down-regulation. Conclusion: To date no clinical study exists in the literature, reporting a poten-tial correlation between IDH and MGMT status and the WNT pathway through the expression of Dkk3.

  1. Introduction

Line 49: Delete "the" prior to "Dkk3".

Line 58: Add "expressed" prior to "enzyme".

Line 63: Revise "founding" to "finding".

Line 69: Change "flowing" to "resulting".

Line 74: Add "the" prior to "cBio".

  • In accordance with the reviewer's suggestions, we have made the appropriate changes in the introduction.

In addition, the activation of the canonical WNT/β-catenin signaling cascade induces the expression of DNA repair enzyme O6-methylguanine-DNA methyltransferase (MGMT) expression, a ubiquitously expressed enzyme that is commonly overexpressed in GB which is associated with resistance to alkylating agents. MGMT is regulated by multiple mechanisms including epigenetic silencing of the MGMT gene by promoter methylation, and is implicated in the development of chemoresistance [17]. In our previous study we investigated “in vitro” and “in vivo” the biological role of Dkk3 in tumor cells of GB, finding that a modulation of this protein could promote apoptosis, providing a potential strategy in the treatment of GB [18]. Particularly, a significant decreasing of Dkk3 was observed in human glioblastoma cell lines, as well as in U-87 MG xenograft tumors and in GB human patient’s tissues, highlighting that a combined modulation of WNT/Dkk3 pathways, simultaneously targeting apoptosis and survival signaling defects, might shift the balance from tumor growth stasis to cytotoxic therapeutic responses, resulting in greater therapeutic benefits. Here, we analyzed the potential prognostic relevance of Dkk3 expression, with immunohistochemistry and western-blot, and its correlation with molecular and histopathological features on 40 GB. In addition, we assessed with statistical analysis, possible correlations between expression of Dkk3 and overall survival, age, gender, Ki-67, p53, MGMT and IDH status. Validation of our data was also considered correlating our results with the cBio Cancer Genomics Portal (cBioPortal) database, creating also a virtual study using data provided by previous studies, and by the samples from 2220 GB patients [19].

  1. Methods

Line 91: Change "mu" (second letter is Greek) to "um" (first letter is Greek)

  • In the text, in the Results section, we have made the requested change.

  1. Results

Table 2: Change "Man" to "Men". Correct "Babinsky" to "Babinski".

Line 209: Revise the title of the Table to "Patient Data".

Line 243: Revise "absence" to "absent".

Line 322: Add "the" prior to "Dkk3".

Line 323: Add "the" prior to "Dkk3".

  • According to your suggestions, in the Results section, we have:

Title changed and table corrected;

Table 2. Patient Data

Clinical Patients Characteristic

Cases

Sex

Women

24

Men

16

Tumor Site

Frontal

15

Temporal

10

Fronto-Insular

6

Temporo-Parietal

6

Multilobar

3

Symptoms

Headache

40

Seizures

14

Personality Changes

10

Increased Intracranial Pressure

10

Nerological Findings

Hemiparesis

23

Central Type Facial Nerve Palsy

14

Babinski Sign

10

Papilledema

8

Unremarkable

6

Surgery

Biopsy

2

Craniotomy: Maximum Resection

12

Craniotomy: Supratotal Resection

26

Adjuvant Therapy

Stupp Protocol

39

Corrected lines 243, 322 and 323.

  • Line 243: with lower or absent expression
  • Lines 322 and 323: Particularly, this analysis highlighted 50% of R132G mutations in GB patients compared to IDH1 wild type observed in all the Dkk3 group, not mutated. The analysis performed on 2220 patients reveled an increase MGMT methylation in the Dkk3 mutated group (23%)
  1. Discussion

Lines 348-349: Delete "resulted, both at" and replace with "in" prior to "immunohistochemistry". Replace "than" with "and" prior to "Western-blot". Add "was" prior to "reduced".

Line 365: Change "reaches" to "reaching".

Line 389: Add "to" prior to "an increase".

Line 400: Replace "confirming" with "supporting".

  • We have corrected the text in accordance with the reviewer's indications.
  • Lines 348-349: In our analyses, Dkk3 expression in immunohistochemistry and Western-blot analyses was reduced or absent in many cases.
  • Line 365: The incidence of GB increases with age, reaching its peak at 75-84 years [3].
  • Line 389: could lead to an increase in survival.
  • Line 400: Dkk3 expression, supporting our hypothesis that Dkk3

Reviewer 2 Report

Comments and Suggestions for Authors

Please find below my comments after reading the revised version of the manuscripts.        

- DKK3 should be capitalized throughout the entire text, and in italic when talking about the gene. Human gene symbols generally are italicized, with all letters in uppercase.

-          Regarding the title and the design of the study: why preliminary results? Is there more ongoing research and the results are not finished yet?

-          When interpreting the results and correlations according the IDH status, it is worth mentioning that in this study we observe only IDH1-132H mutated patients. so, when generalizing the results, the authors should be very careful about this point. Maybe the other mutations can show different results! At least, the authors should be aware of this important limitation and state it. This is very important.

-          In my point of view, I still believe that the paper needs more English editing.

-          Discussion, line 348-349: the idea is not clear due to the English style. Please rephrase.

-          Line 350: “suggesting a role on tumor growth 350 of Dkk3”, probably change to “suggestion a role for DKK3 in promoting tumor growth” ?

-          Line 351:  tumor suppressor gene instead of “tumor-suppression-gene”

-          Line 352: increased instead of increasing?

-          Line 357 : “compared to” rather than “instead of” ?

-          Line 381-382: “For the first time, in this study, we demonstrated that both patients with IDH wild- 381 type and IDH mutated showed a moderate and high expression of Dkk3, compared to the 382 remaining patients (6) with absent or low Dkk3 expression.” The sentence is not clear. The second half of the sentence does not make any sense. Please rephrase and explain.

Comments on the Quality of English Language

English editing is still needed. 

Author Response

RE: “Dkk3 Expression in Glioblastoma: Correlations with Biomolecular Markers. Preliminary Results

Manuscript ID: ijms-2907354

Response to your comments

Reviewer #2

We thank Reviewer for suggestions. We have read reviewer’s comments, and have revised our manuscript accordingly.

Comments and Suggestions for Authors

  • DKK3 should be capitalized throughout the entire text, and in italic when talking about the gene. Human gene symbols generally are italicized, with all letters in uppercase.
  • In accordance with the reviewer's observation we have corrected the text.

  • Regarding the title and the design of the study: why preliminary results? Is there more ongoing research and the results are not finished yet?
  • We thank the reviewer for his comment, which gives us the possibility to better explain our data. Really, we already published two manuscripts concerning the role of Dkk-3 in meningiomas and in gliomas (this latter a pre-clinical study) (Casili G et al., TLR-4/Wnt Modulation as New Therapeutic Strategy in the Treatment of Glioblastomas. Oncotarget 2018, 9, 37564-37580; Caffo M et al. Modulation of Dkk-3 and Claudin-5 as New Therapeutic Strategy in the Treatment of Meningiomas. Oncotarget 2017, 8, 68280-68290). However, we thought to define “preliminary results”, because we wanted to be cautious in making definitive conclusions on the interpretation of the protein data of this pathway, which is little studied in gliomas. The term "preliminary" did not refer to ongoing research, although we hope for multicenter studies that involve the study of the entire WNT pathway, collecting a higher number of patients. However, according to your suggestion we modified the title.

  • When interpreting the results and correlations according the IDH status, it is worth mentioning that in this study we observe only IDH1-132H mutated patients. so, when generalizing the results, the authors should be very careful about this point. Maybe the other mutations can show different results! At least, the authors should be aware of this important limitation and state it. This is very important.
  • We thank the reviewer for his suggestion, which gives us the opportunity to better specify this point. In a dataset of 40 patients, having 8 cases of grade IV astrocytoma IDH mutated is in line with what has recently been reported in the literature (Ostrom QT et al. CBTRUS Statistical Report: primary brain and other central nervous system tumors diagnosed in the United States in 2014-2018. Neuro Oncol. 2021; 23(Suppl 2). In addition, we also noted some mistakes and corrected accordingly the table 1, including all the IDH 1 mutations found in our 8 patients (R132H, R132C, R132X).

Table 1. Biomolecular data and survival of each patient.

Patient

Dkk3

Ki67 (%)

IDH1

MGMT promoter Methylation (%)

P53

Survival (months)

1

2

40

WT

6

+

9

2

0

20

WT

8

-

2

3

2

30

WT

6

+

47

4

0

40

WT

20

+

11

5

0

80

WT

8

+

1

6

2

40

WT

6

+

7

7

2

10

WT

8

-

1

8

2

50

WT

8

-

3

9

2

10

R132H

45

+

54

10

2

80

R132C

45

+

23

11

1

>20

WT

6

-

15

12

2

>50

WT

30

+

1

13

1

>20

WT

55

+

22

14

2

>50

WT

6

+

19

15

1

20

WT

55

-

44

16

2

25

WT

7

+

4

17

2

20

WT

4

+

13

18

0

10

WT

27

-

28

19

1

>35

R132H

4

+

19

20

2

>50

WT

4

+

14

21

1

10

WT

30

+

8

22

1

20

WT

25

+

1

23

2

15

WT

10

+

43

24

0

50

WT

40

+

10

25

2

15

WT

8

+

1

26

2

10

WT

6

+

6

27

1

20

R132H

25

+

2

28

2

10

R132H

10

+

2

29

2

15

WT

6

+

50

30

2

10

WT

5

+

21

31

1

30

WT

12

+

16

32

2

10

WT

20

-

3

33

2

15

R132X

8

+

20

34

2

10

R132H

10

+

21

35

2

10

WT

10

+

43

36

2

20

WT

8

+

2

37

2

10

WT

10

+

11

38

1

30

WT

30

+

27

39

0

80

WT

55

-

18

40

2

20

R132H

10

+

13

We correlated the number of patients both mutated than IDH wild type with DKK3 expression and survival. Therefore, we also modified the text as follows:

Result section (line 267-269): "an increased survival, expressed in months, was observed in patients with moderate and high DKK3 expression, both with IDH wild type and mutated" (Fig. 7B)”.

Discussion section (line 380-385): “Major advances in cancer genetics have revealed that the genes encoding IDHs are frequently mutated in a variety of human malignancies, including GB [30-31]. More than 90% of the mutations in IDH1 are R132H that has been associated with significantly improved prognosis and longer progression-free and overall survival [32-33]. IDH1-R132H mutation leads to both a less aggressive phenotype and radiosensitization of glioma cells [34]”.

  • Discussion, line 348-349: the idea is not clear due to the English style. Please rephrase.
  • Line 350: “suggesting a role on tumor growth of Dkk3”, probably change to “suggestion a role for DKK3 in promoting tumor growth”?
  • Line 351: tumor suppressor gene instead of “tumor-suppression-gene”
  • Line 352: increased instead of increasing?
  • Line 357: “compared to” rather than “instead of”?
  • In accordance with the reviewer's suggestions, we have made the appropriate changes in the revised text.

  • Line 381-382: “For the first time, in this study, we demonstrated that both patients with IDH wild- type and IDH mutated showed a moderate and high expression of Dkk3, compared to the remaining patients (6) with absent or low Dkk3 expression.” The sentence is not clear. The second half of the sentence does not make any sense. Please rephrase and explain.
  • We thank the reviewer for his precious comment, giving us the possibility to better explain our concepts. Really, we would like to specify that this our is the first manuscript in which a correlation between IDH status in patients affected by GB wild type and grade IV mutated astrocytoma and DKK3 expression has been reported. Therefore, accordingly, we modified and rephrased the sentence.

“For the first time, in this study, we have correlated the IDH status with Dkk-3 expression. Our results showed a moderate and high expression of DKK3, in 34 patients (both wild-type and IDH mutated) comparing to the remaining patients (6) with absent or low DKK3 expression.”